# Joint observation in NICU (JOIN): A randomized controlled trial testing an early, one-session intervention during preterm care to improve perceived maternal self-efficacy and other mental health outcomes

Juliane Schneider[1,2]*, Mathilde Morisod Harari[3], Noémie Faure[4,5], Alain Lacroix[6], Ayala Borghini[7], Jean-François Tolsa[1], Antje Horsch[1,6], on behalf of the JOIN Research Consortium[¶]

1 Department of Woman-Mother-Child, Clinic of Neonatology, Lausanne University Hospital and University of Lausanne, Lausanne, Switzerland, 2 The Sense, Innovation, and Research Center, Lausanne, Switzerland, 3 Division of Child and Adolescent Psychiatry, Department of Psychiatry, Lausanne University Hospital and University of Lausanne, Lausanne, Switzerland, 4 Centre Sages-Femmes, Vevey, Switzerland, 5 UniVers Famille, Châtel-St-Denis, Switzerland, 6 Institute of Higher Education and Research in Healthcare, University of Lausanne, Lausanne, Switzerland, 7 Haute école de travail social, Geneva, Switzerland

¶ The membership list of the JOIN Research Consortium is provided in the Acknowledgments.
* Juliane.Schneider@chuv.ch

**Data Availability Statement:** There are legal restrictions on sharing de-identified data set

## Abstract

### Background

Parents of preterm infants in the Neonatal Intensive Care Unit (NICU) environment may experience psychological distress, decreased perceived self-efficacy, and/or difficulties in establishing an adaptive parent-infant relationship. Early developmental care interventions to support the parental role and infant development are essential and their impact can be assessed by an improvement of parental self-efficacy perception. The aims were to assess the effects of an early intervention provided in the NICU (the Joint Observation) on maternal perceived self-efficacy compared to controls (primary outcome) and to compare maternal mental health measures (perceived stress, anxiety, and depression), perception of the parent-infant relationship, and maternal responsiveness (secondary outcomes).

### Methods

This study was a monocentric randomized controlled trial registered in clinicatrials.gov (NCT02736136), which aimed at testing a behavioural intervention compared with treatment-as-usual. Mothers of preterm neonates born 28 to 32 6/7 weeks gestation were randomly allocated to either the intervention or the control groups. Outcome measures consisted of self-report questionnaires completed by the mothers at 1 and 6 months after enrollment and assessing perceived self-efficacy, mental health, perception of the parent-infant relationship and responsiveness, as well as satisfaction with the intervention.

because no consent to do so is available from the participants according to the protocol submttted to the ethics committee in 2015.The data underlying the results presented in the study are available from the Institutional Review Board by contacting Mrs. Jeanne-Pascale Simon (Jeanne-Pascale. Simon@chuv.ch).

**Funding:** The authors received no specific funding for this work.

**Competing interests:** The authors have declared that no competing interests exist.

## Results

No statistically significant group effects were observed for perceived maternal self-efficacy or the secondary outcomes. Over time, perceived maternal self-efficacy increased for mothers in both groups, while anxiety and depression symptoms decreased. High satisfaction with the intervention was reported.

## Conclusions

The joint observation was not associated with improved perceived maternal self-efficacy or other mental health outcomes, but may constitute an additional supportive measure offered to parents in a vulnerable situation during the NICU stay.

## Introduction

Prematurity and consecutive neonatal complications, as well as exposure to the Neonatal Intensive Care Unit (NICU) environment may affect the developmental trajectory of the preterm brain, which is vulnerable to insults and altered maturation [1]. Besides, the peculiar birth circumstances can enhance parental vulnerability in the construction of their parental role and the parent-infant relationship [2]. These different factors may interact together to influence the infant's short- and long-term neurodevelopment.

With the aim of improving neurological outcomes, the developmental care concept was introduced in the NICU in the 1990s [3]. Various programs were developed, essentially following three main goals [4]; the first consists of reducing abnormal sensory stimulation in the NICU, by decreasing exposure to sound, light, pain, and stress [5, 6]. The second goal aims to adapt the NICU environment to provide more physiological stimulation (tactile, auditory, visual, vestibular), promoting regulation and well-being by individualizing care to the preterm neonate [7]. The third aim focuses on supporting parents' engagement toward their neonate and on helping to build a healthy relationship despite physical and emotional barriers [8].

Parents of preterm infants may experience high levels of stress and demonstrate a broad range of coping resources. The origin of parental stress is multifactorial and is generated namely by the fear of losing their infant, his/her critical medical condition, the perception of extreme fragility, the difficulties in adaptation to the NICU environment, and the physical separation with the neonate [9]. Mental health issues may develop or latent signs may be exacerbated, in the form of anxiety, depression or post-traumatic stress disorder, with persisting effects in the first years after birth [10, 11]. Previous research suggested that parental stress and psychological vulnerability interfere with the infant's socio-emotional and cognitive development [12]. Many factors may disrupt the construction and maturation of the parent-infant relationship throughout the hospitalization, placing a risk on bonding and secure attachment [13]. High sensitivity and responsiveness foster engagement in interactive and protective behaviors toward the neonate, but these conditions require parental psycho-emotional availability and compensated mental health issues [14]. On the other hand, preterm neonates are prone to dysregulated behaviors and decreased capacity of auto-regulation, making their communication cues more subtle and difficult to interpret [15]. Altogether, capacities of co-regulation and synchrony of the parent-neonate dyad may be durably and negatively affected.

The concept of perceived parental self-efficacy is defined as "beliefs or judgements a parent holds of their capabilities to organize and execute a set of tasks related to parenting a child",

and needs to be distinguished from parental confidence and parental competence [16, 17]. The latter refers to the conviction that others hold about the parent's abilities to achieve something. In that sense, the parents may possess the skills but are not necessarily able to integrate them in a specific task. On the other hand, self-confidence refers to the belief in one's own abilities and judgements and constitutes a stable state of certainty which is not situation-dependent. While relying on self-confidence, self-efficacy more precisely relates to the conviction to succeed in actively learning new skills and accomplishing specific tasks (e.g., the mother's ability to feed or soothe her baby), while adapting to unfamiliar situations [16]. In addition, perceived parental self-efficacy mediates the relationship between psycho-social risk factors and maternal competences [18]. Yet, previous studies reported associations between low perceived self-efficacy and parental depression [18], high levels of parenting stress [19], low family support, and difficult infant temperament [20]. Conversely, parents reporting high-perceived self-efficacy demonstrated more sensitive and responsive behaviours, which in turn was related to better infant socio-emotional outcomes [21].

Assuming that the perception of parental self-efficacy may be modulated by means of the interaction with health professionals, we developed a targeted intervention, the joint observation (JOIN: Joint Observation in Neonatology), which was designed according to the global approach of family-centered developmental care. So far, very few interventions have focused on enhancing perceived parental self-efficacy [22, 23]. The joint observation, previously described in details elsewhere [24, 25], appeared to enhance the quality of the mother-infant interactions [25]. This intervention is provided by an interdisciplinary partnership of professionals, including NICU nurses, paediatricians, clinical child psychologists, and child psychiatrists. It is based upon four theories of neonatal and infant development: 1) detection of the neonate's competences, stress cues, and fragilities to adjust to his/her regulation needs, as proposed by Brazelton and Nugent [26]; 2) individualisation of care to avoid overstimulation and to support self-regulation and competences, according to the synactive model of Als [27]; 3) sensori-motor approach developed by Bullinger, which assesses sensory dystimulations and supports the management of tonico-postural disturbances and treatment of multisensory information [28]; 4) interactive guidance, which uses video feedback to analyse the parent-infant interactions, allowing the demonstration of competences and resources of both the parent and the neonate [29, 30]. In this intervention, the professionals seek to highlight the adaptive capacity, the competences, and the interactive signals of the neonate, and to enhance parental sensitivity and responsiveness, pointing out positive interactions and phases of mutual adjustment.

Based on previous data [25], the first objective of this randomized controlled trial (RCT) was to assess the effects of a behavioural intervention provided at an early stage in the NICU (the Joint Observation) on maternal perceived self-efficacy one month after study enrolment. Secondary objectives were to compare in the two groups measures of maternal mental health (perceived stress, anxiety, and depression), perception of the parent-infant relationship, and maternal responsiveness. We hypothesized that mothers exposed to the intervention would report higher perceived self-efficacy and improved mental health measures compared to control participants.

## Materials and methods

The protocol for the present study has been published and described extensively elsewhere [24]. The local ethical committee ('Commission cantonale d'éthique de la recherche sur l'être humain, Vaud'—CER-VD) approved the study protocol (# 496/15) on March 1st, 2016. This RCT was registered in clinicatrials.gov (NCT02736136). This manuscript follows the CONSORT guidelines.

## Participants

During a 48-month period (March 2016 to February 2020), mothers of preterm neonates born between 28 and 32 6/7 weeks of gestational age (GA) hospitalized in the level III NICU of a Swiss University Hospital and aged less than 8 weeks of life were eligible for inclusion in this RCT. Exclusion criteria regarding the mother were: age less than 18 years, established intellectual disability or psychotic illness, insufficient local language skills level to complete questionnaires. Exclusion criteria regarding the preterm neonate was related to cardiorespiratory instability (severe brady-apnoea syndrome, oxygen requirement >30%) to ensure survival during the study period. Recruitment was performed by the study nurses.

## Sample size calculation *(for further detail, see* [24]*)*

The power calculation was based on previous publications measuring perceived parental self-efficacy in parents of term [18] and preterm [31] neonates. As no previous study compared this specific outcome after a similar intervention in two groups of mothers of preterm neonates in the NICU, we made the assumption that the mothers of preterm neonates benefitting from the JOIN intervention will report comparable perceived self-efficacy as mothers of term neonates. Using the G*Power software [32] allowing sample size determination and according to the published means and SD in these two samples (term: M = 65.9, SD = 8.2; preterm: M = 58.1, SD 12.57), we needed to recruit 68 participants ($\alpha$ = 0.05, 1-$\beta$ = 0.80, unilateral hypothesis). To anticipate possible withdrawal, we planned to enroll 80 mother-infant dyads in the present study.

## Trial design, procedure, data collection and timing

This monocentric RCT aimed to test an intervention compared with treatment-as-usual. The recruitment of mothers of preterm neonates was performed by research nurses who approached the eligible participants once their infants were stable enough to avoid disturbance during a critical period and ensure their emotional availability. Using a computer-generated list of random blocks (https://www.sealedenvelope.com), a research assistant generated the random allocation sequence, which was concealed from the principal investigator in sequentially numbered, opaque, and sealed envelopes. Two groups of participants were created on the basis of a randomisation 1:1. The envelopes were opened after the signature of the consent and the completion of the baseline questionnaires (for further details see [24]).

   The principal investigator and the statistician were blinded to group allocation. As the intervention took place in the clinical context of the NICU and no placebo intervention was performed, the clinical team administering the intervention was de facto not blinded for group allocation but did not participate in subsequent data collection and follow-up. Similarly, no blinding was possible for the participants.

## Control group

After giving written consent, participants of the control group were asked to complete a set of questionnaires at three time points : T1 at baseline (recruitment), T2 at 1-month after enrollment, and T3 at 6 months corrected-age of the infant (CA). The mother-infant dyads received treatment-as-usual in the NICU after a preterm birth.

## Intervention group

Mothers assigned to the intervention group were asked to complete the questionnaires at the three time points defined previously. The intervention was performed after enrollment and completion of the baseline questionnaires once the infant's clinical state was stabilized.

The early, one-session intervention, the joint observation, consisted of two phases that took place on the same day: firstly, the child psychiatrist or psychologist and the NICU nurse, called the observers, jointly observed a period of care administered to the neonate jointly by her mother and a NICU nurse [24], approximately for 30 min. Observers did not intervene during the care procedure. This period of care was video-recorded in order to conduct the second phase of the intervention : a video-feedback session, based on interactive guidance, in the presence of the observers and both the mother and the NICU nurse. The discussion was illustrated by several short extracts of the previous recorded care procedure (4–6 extracts of 10-30s each), carefully selected by the observers just prior to the video-feedback session. This session lasted approximately for 60 min.

At the end of the intervention, mothers were asked to fill in a questionnaire assessing their satisfaction related to the intervention. For more details about the joint observation, see [24].

## Measures : Self-report questionnaires on maternal mental health

**Primary outcom.** *Perceived Maternal Parenting Self-efficacy (PMP-SE) [31]*. This 20-items questionnaire includes four subscales (care taking procedure, evoking behaviours, reading behaviours/signalling, situational beliefs) specifically designed to assess perception of parental ability for mothers experiencing NICU hospitalization of their preterm neonates. The questionnaire was developed in a population of relatively healthy preterm neonates and authors warn of the need of a cross-cultural validation for a more general application outside Europe. Examples of questions: Subscale 1 « I am good at changing my baby », Subscale 2 « I can make my baby calm when he/she has been crying », Subscale 3 « I am good at understanding what my baby wants », Subscale 4 « I believe that my baby responds well to me ». Responses on each item are coded on a 4-points Likert scale (*1 = strongly disagree, 2 = disagree, 3 = agree, 4 = strongly disagree*). The scores range from 20–80, with a higher score indicating higher perceived maternal parenting self-efficacy. A forward-backward method [33] was used to obtain a valid French version of the questionnaire.

**Secondary outcomes.** *Parental Stressor Scale : Neonatal Intensive Care Unit (F-PSS-NICU) [34]*. This 31 item-questionnaire with good psychometric properties assesses the parental stress specifically in NICU in three domains: impact of the environment, behaviour and aspect (e.g., respiratory pattern, colour, equipment, skin lesions) of the infant, and parental role. This questionnaire was 'designed to measure parental perception of stressors arising from the physical and psychosocial environment of the NICU' [34]. The total score ranges from 31 to 155, with a higher score indicating higher exposure to stressors in the NICU environment.

*Parenting Stress Index–Short Form (PSI—SF) [35]*. This questionnaire is the shortened version of the PSI and assesses stress related to parenthood during the first three years of the child. The 36 items are divided into three subscales: parental distress, parent-infant dysfunctional interaction, and child difficulties. A higher score indicates higher level of stress related to parenting. This questionnaire has been validated in French and displays good psychometric properties in studies involving parents of children born preterm with various degree of prematurity and medical risk and issued from a racially and socio-economically diverse population [36].

*Hospital Anxiety and Depression Scale (HADS)*. This 14-item questionnaire assesses the severity of symptoms of anxiety and depression. The total score ranges from 0–52, with a higher score indicating a higher level of psychological distress. The 7-item subscales for anxiety or depressive symptoms range from 0–26, with a higher score indicating higher levels of symptoms. While the original questionnaire was not developed to specifically assess mental health of parents after preterm birth, Vriend et al. recently reported increased level of anxiety and depression in parents of preterm neonates with social risks (migration background,

educational level and employment status) [37]. The validated French version has good psycho-metric characteristics [38].

*Edinburgh Postnatal Depression Scale (EPDS).* Maternal depression was also assessed with the Edinburgh Postnatal Depression Scale, a 10-item questionnaire with a 4-points Likert-scale. This scale focuses on the symptoms experienced over the last seven days, with a higher score indicating more symptoms of depression. Although this questionnaire is not intended to specifically evaluate depressive symptoms after a preterm birth, multiple studies reported these outcomes in a population of mothers of preterm infants [39, 40]. This questionnaire has been validated in French and displays good psychometric properties [41].

*Mother-to-Infant Bonding Scale (MIBS) [42].* The mother-infant bonding is assessed with eight sentences rated by mothers on a 5-points Likert scale (e.g., « I feel protective towards my child »). A higher total score indicates more bonding problems. The questionnaire does not specifically assess bonding after NICU hospitalization, but our group reported this outcome in mothers of term neonates admitted after neonatal asphyxia [43]. The questionnaire was trans-lated into French and demonstrates good psychometric characteristics.

*Infant Behaviour Questionnaire–Revised Very Short Form (IBQ-R) [44].* This 37-item ques-tionnaire assesses the temperament of the infant based on parental report. Parent rates each item on a 7-points Likert-scale regarding the frequency of the behaviour during the previous two weeks. The total score ranges from 37–259, with a higher score indicating a more challeng-ing infant temperament. This questionnaire previously proved to reliably assess temperament in former preterm children during the first year of age [45]. The French version has been vali-dated and shows good psychometric properties.

*Modified Medical Outcomes Study Social Support Survey (m-MOS-SS) [46].* This question-naire assesses different aspects of social support on four scales, with a higher score indicating the more frequent availability of social support. A forward-backward method was used to obtain a valid French version of the questionnaire [33].

*Questionnaire of satisfaction and acceptability of the intervention.* This questionnaire, com-pleted at the end of the intervention by mothers, covers general questions on satisfaction regarding the intervention. In addition, mothers filled in six questions on its setting, value and usefulness, and three questions focusing on the acceptability of the intervention.

## Statistical analysis

All analyses were carried out with RStudio v1.3.1093 running R v4.1.0. We disregarded partici-pants having not replied to any items in questionnaires (S1 Table). We imputed questionnaires missing at random values per time point and at item level using the median value of the partici-pant items. Four participants were not assigned to a group and thus excluded from the analy-sis. The flow chart of the study is available in Fig 1.

As the outcomes, resulting from the aggregation of Likert scale type of items, were some-time deviating from normal distributions we tried to enforce non-parametric tests where pos-sible. To analyze the between-group differences at post intervention and 6 months, outcomes baselines as well as the potential confounding variables were included in covariance analyses, where applicable. These included the age of the mother, the gender of the baby, the socio-eco-nomic status [47], and number of children. For the primary outcome analyses, the mMOS-SS score at baseline was included in the covariance analysis as well, as this correlated statistically significantly with the primary outcome at baseline. The effect sizes and the corresponding sta-tistical power were estimated at post intervention and 6 months.

We split the 2x3 factorial design into two simple within-subject (repeated measures) designs to analyze the within-group differences at pre-intervention, post-intervention, and 6 months.

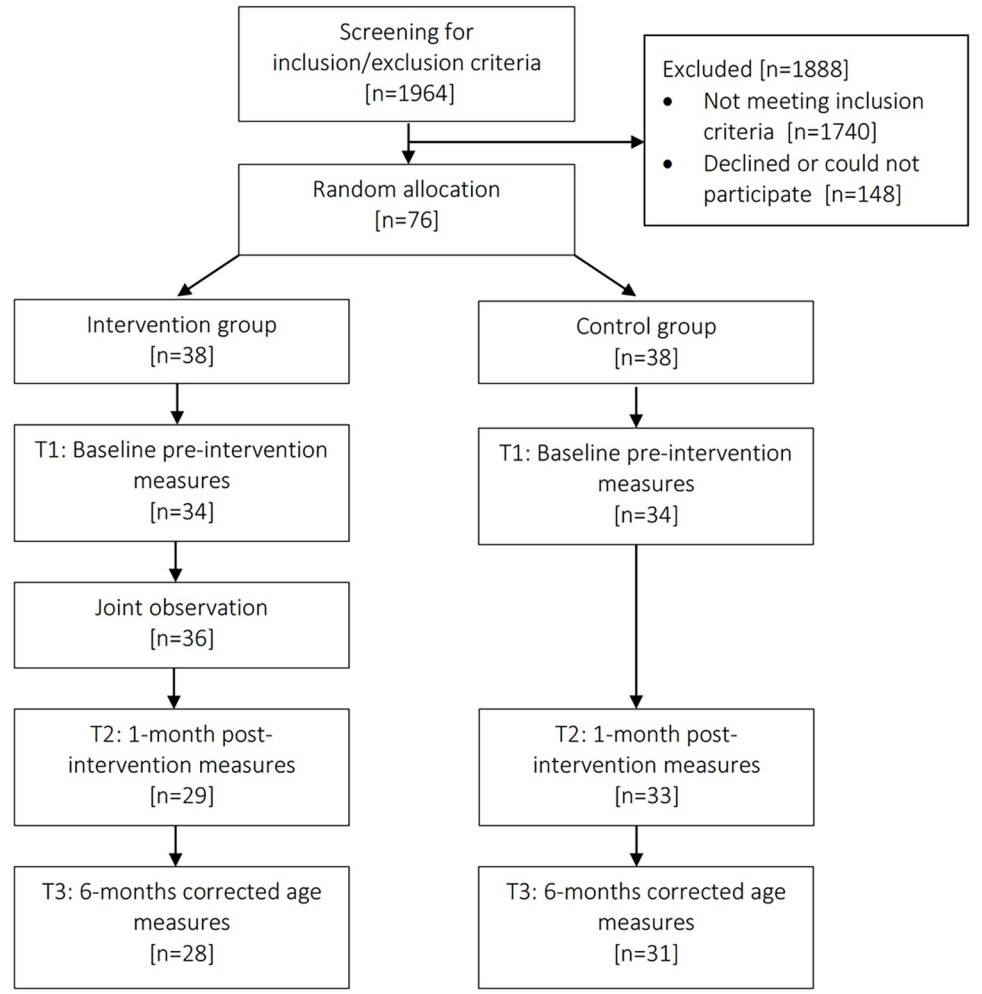

**Fig 1. Flow chart of the study.**

To consider the repeated measure structure we used Aligned Ranks Transform (ART) models when there was no covariate statistically significantly related with the outcome and Linear Mixed Effect (LME) models otherwise. For each group, the effect size and related statistical power were estimated at pre-intervention, post-intervention, and 6-month.

## Results

The group characteristics at baseline (pre-intervention) are summarized in Table 1 and S2 Table. All variables looked well balanced between groups.

All questionnaires showed a fair to good internal consistency (i.e., between 0.70 and 0.95) at pre-intervention, post-intervention and at 6 months, except for the following questionnaires: the IBQ-R Surgency at pre-intervention ($\alpha = 0.598$, 95% CI [0.523, 0.723]) and 6 months ($\alpha = 0.641$, 95% CI [0.527, 0.716]; the IBQ-R Effortful Control at pre-intervention ($\alpha = 0.690$, 95% CI [0.611, 0.742] and at 6 months ($\alpha = 0.690$, 95% CI [0.590, 0.739]; and the MIBS at pre-intervention ($\alpha = 0.614$, 95% CI [0.526, 0.673] and post intervention ($\alpha = 0.639$, 95% CI [0.528, 0.662]) (see S3 Table).

**Table 1. Between-group differences at baseline—Maternal and infant sociodemographic variables.**

| Mother | Intervention N = 37 | Control N = 36 |
|---|---|---|
| Age (M, SD) | 31.78 (4.71) | 32.61 (5.19) |
| Education (N, %) | | |
| 1: Primary school | 1 (1.37) | 0 (0) |
| 2: Middle school | 1 (1.37) | 3 (4.11) |
| 3: Secondary/high school | 5 (6.85) | 2 (2.74) |
| 4: Apprenticeship | 7 (9.59) | 8 (10.96) |
| 3 + 4: Both | 0 (0) | 1 (1.37) |
| 5: University | 22 (30.14) | 22 (30.14) |
| Missing | 1 (1.37) | 0 (0) |
| Migrant (yes, %) | 13 (17.81) | 8 (10.96) |
| Missing | 0 (0) | 1 (1.37) |
| Marital status (N, %) | | |
| 1: Single | 11 (15.07) | 6 (8.22) |
| 2: Married | 25 (34.25) | 23 (31.51) |
| 3: Separated | 0 (0) | 1 (1.37) |
| 4: Divorced | 1 (1.37) | 2 (2.74) |
| 6: Other | 0 (0) | 3 (4.11) |
| Missing | 0 (0) | 1 (1.37) |
| Number of pregnancies (M, SD) | 1 (1.68) | 1.06 (1.16) |
| Missing | 0 | 1 |
| Number of children (N, %) | | |
| 0 | 4 (5.48) | 2 (2.74) |
| 1 | 20 (27.40) | 18 (24.66) |
| 2 | 9 (12.33) | 9 (12.33) |
| 3 | 0 (0) | 7 (9.59) |
| 4 | 4 (4.58) | 0 (0) |
| Largo score (N, %) | | |
| 1: | 5 (6.85) | 10 (13.70) |
| 2: | 17 (23.29) | 12 (16.44) |
| 3: | 6 (8.22) | 8 (19.96) |
| 4: | 7 (9.59) | 3 (4.11) |
| 5: | 1 (1.37) | 3 (4.11) |
| 6: | 1 (1.37) | 0 (0) |
| Infant | | |
| Gestational Age (M, SD) | 30.05 (1.44) | 30.39 (1.26) |
| Sex (female, %) | 20 (27.40) | 17 (23.29) |
| Weight (grams) | 1295.38 (354.19) | 1296.06 (401.42) |
| Apgar score at 5 min | 8.68 (1.58) | 8.46 (1.20) |

Note: The Largo score evaluates the socio-economic status of the mother, according to her educational level. The score of 1 represents university graduation and the score of 6, completion of primary school.

The between-group analyses did not show any statistically significant group effect, neither at post-intervention, nor at 6 months, for the variables in scope (see results in Table 2). Controlling for covariates did not change these results (results not shown). The results shown that all the between-group analyses were underpowered, due to small effect sizes combined with

**Table 2. Between-group differences at post-intervention and 6 months.**

| Outcomes | Group | | | |
|---|---|---|---|---|
| Time point | Intervention (N, Med) | Control (N, Med) | Group effect | Effect size (eta2, 95%CI, magnitude, power) |
| **PMP-SE** | | | | |
| Post-Int. | 26, 66 | 28, 69.5 | F(1,50) = 0.26, p = .61 | η2 = 0.0052 [0,1] (small), power = 0.081 |
| 6-Mths | 25, 72 | 26, 74.5 | F(1,47) = 0.19, p = .66 | η2 = 0.0041 [0,1] (small), power = 0.072 |
| **PSI Total** | | | | |
| Post-Int. | 27, 148 | 30, 150.5 | F(1,54) = 0, p = .96 | η2 = 1e-04 [0,1] (small), power = 0.05 |
| 6-Mths | 24, 145 | 27, 157 | F(1,48) = 0.4, p = .53 | η2 = 0.0083 [0,1] (small), power = 0.097 |
| **PSI-Parental Distress** | | | | |
| Post-Int. | 29, 46 | 31, 49 | F(1,57) = 0.02, p = .9 | η2 = 3e-04 [0,1] (small), power = 0.052 |
| 6-Mths | 26, 48 | 28, 50.5 | F(1,51) = 0.01, p = .94 | η2 = 1e-04 [0,1] (small), power = 0.051 |
| **PSI-Parent-Child Dysfunctional Interaction** | | | | |
| Post-Int. | 33, 52.5 | 28, 54 | F(1,58) = 0.72, p = .4 | η2 = 0.0123 [0,1] (small), power = 0.136 |
| 6-Mths | 24, 53 | 23, 55 | F(1,44) = 0.71, p = .41 | η2 = 0.0158 [0,1] (small), power = 0.134 |
| **PSI-Difficult Child** | | | | |
| Post-Int. | 27, 54 | 29, 54 | F(1,47) = 0.02, p = .89 | η2 = 4e-04 [0,1] (small), power = 0.052 |
| 6-Mths | 24, 53 | 26, 55 | F(1,41) = 0.73, p = .4 | η2 = 0.0175 [0,1] (small), power = 0.137 |
| **HADS Total** | | | | |
| Post-Int. | 29, 18 | 31, 14 | F(1,57) = 1.23, p = .27 | η2 = 0.0211 [0,1] (small), power = 0.199 |
| 6-Mths | 27, 12 | 30, 8.5 | F(1,54) = 0.87, p = .36 | η2 = 0.0158 [0,1] (small), power = 0.154 |
| **HADS-Anxiety** | | | | |
| Post-Int. | 29, 9 | 31, 7 | F(1,57) = 0.7, p = .41 | η2 = 0.0121 [0,1] (small), power = 0.133 |
| 6-Mths | 27, 6 | 30, 6 | F(1,54) = 0.57, p = .45 | η2 = 0.0105 [0,1] (small), power = 0.118 |
| **HADS-Depression** | | | | |
| Post-Int. | 28, 8 | 31, 6 | F(1,56) = 0.95, p = .33 | η2 = 0.0168 [0,1] (small), power = 0.165 |
| 6-Mths | 27, 5 | 30, 3.5 | F(1,54) = 0.68, p = .41 | η2 = 0.0125 [0,1] (small), power = 0.131 |
| **F-PSS-NICU Total** | | | | |
| Post-Int. | 28, 2.98 | 31, 3.16 | F(1,51) = 0.09, p = .76 | η2 = 0.0018 [0,1] (small), power = 0.061 |
| 6-Mths | 24, 2.98 | 28, 3.19 | F(1,44) = 0.13, p = .72 | η2 = 0.0029 [0,1] (small), power = 0.065 |
| **PSS-Visual and Auditive** | | | | |
| Post-Int. | 28, 3 | 31, 2.78 | F(1,56) = 0.2, p = .66 | η2 = 0.0035 [0,1] (small), power = 0.073 |
| 6-Mths | 27, 3 | 29, 2.78 | F(1,53) = 0.06, p = .8 | η2 = 0.0012 [0,1] (small), power = 0.057 |
| **PSS-Baby Behavior** | | | | |
| Post-Int. | 30, 2.77 | 32, 3.08 | 0.58, p = .446 | -0.006 [-0.01, 0.04] (small) |
| 6-Mths | 25, 2.62 | 27, 2.92 | 0.24, p = .627 | -0.01 [-0.01, 0.07] (small) |
| **PSS-Parent Role** | | | | |
| Post-Int. | 29, 2.69 | 30, 3.19 | F(1,56) = 0.77, p = .39 | η2 = 0.0135 [0,1] (small), power = 0.141 |
| 6-Mths | 23, 2.62 | 25, 2.92 | F(1,45) = 0.03, p = .87 | η2 = 6e-04 [0,1] (small), power = 0.053 |
| **IBQ-R Total** | | | | |
| Post-Int. | 29, 1.78 | 30, 2.18 | F(1,56) = 0.01, p = .92 | η2 = 2e-04 [0,1] (small), power = 0.051 |
| 6-Mths | 25, 3.92 | 28, 4.22 | F(1,50) = 0.94, p = .34 | η2 = 0.0184 [0,1] (small), power = 0.162 |
| **IBQ-R Surgency** | | | | |
| Post-Int. | 27, 1.15 | 30, 1.19 | F(1,53) = 0.01, p = .91 | η2 = 2e-04 [0,1] (small), power = 0.051 |
| 6-Mths | 25, 4.46 | 26, 4.54 | F(1,47) = 0.86, p = .36 | η2 = 0.018 [0,1] (small), power = 0.153 |
| **IBQ-R Negative Affect** | | | | |
| Post-Int. | 27, 1.42 | 30, 2.17 | F(1,54) = 2.76, p = .1 | η2 = 0.0487 [0,1] (small), power = 0.383 |

*(Continued)*

**Table 2.** (Continued)

| Outcomes | Group | | | |
|---|---|---|---|---|
| Time point | Intervention (N, Med) | Control (N, Med) | Group effect | Effect size (eta2, 95%CI, magnitude, power) |
| 6-Mths | 25, 2.5 | 28, 2.79 | $F_{(1,50)} = 0.58$, $p = .45$ | $\eta2 = 0.0114$ [0,1] (small), power = 0.118 |
| **IBQ-R Effortful Control** | | | | |
| Post-Int. | 29, 2.75 | 30, 3.25 | $F_{(1,56)} = 0.04$, $p = .84$ | $\eta2 = 8e\text{-}04$ [0,1] (small), power = 0.055 |
| 6-Mths | 25, 4.83 | 28, 5.42 | $F_{(1,50)} = 1.41$, $p = .24$ | $\eta2 = 0.0274$ [0,1] (small), power = 0.221 |
| **MIBS** | | | | |
| Post-Int. | 27, 2 | 30, 2 | $F_{(1,48)} = 0.08$, $p = .78$ | $\eta2 = 0.0016$ [0,1] (small), power = 0.059 |
| 6-Mths | 25, 1 | 28, 3 | $F_{(1,45)} = 0.3$, $p = .59$ | $\eta2 = 0.0066$ [0,1] (small), power = 0.085 |
| **EPDS** | | | | |
| Post-Int. | 29, 9 | 31, 7 | $F_{(1,57)} = 0.05$, $p = .83$ | $\eta2 = 8e\text{-}04$ [0,1] (small), power = 0.055 |
| 6-Mths | 26, 7.5 | 29, 4 | $F_{(1,52)} = 1.51$, $p = .22$ | $\eta2 = 0.0282$ [0,1] (small), power = 0.233 |

Abbreviations: EPDS: Edinburgh Postnatal Depression Scale; F-PSS-NICU: Parental Stressor Scale: neonatal intensive care unit; HADS: Hospital Anxiety and Depression Scale; IBQ-R: Infant Behavior Questionnaire-Revised Very Short Form; MIBS: Mother-to-Infant Bonding Scale; m-MOS-SS: Modified Medical Outcomes Study Social Support Survey; PMP-SE: Perceived Maternal Self-efficacy; PSI: Parenting Stress Index.

Notes: Effect size interpretation, as suggested by Cohen (1988): small ($\eta2 \geq 0.01$), medium ($\eta2 \geq 0.06$), and large ($\eta2 \geq 0.14$)

undersized samples, increasing the likelihood of a type 2 error. The results of the within-group group analyses presented below are detailed in Table 3.

For the **Perceived Maternal Parenting Self-Efficacy (PMP-SE)**, statistically significant time effects were found (S1 Fig), for both the intervention group and for the control group. For the intervention group, a) the average PMP-SE score at post-intervention was statistically significantly higher than at pre-intervention; b) the average PMP-SE score at 6 months was statistically significantly higher than at pre-intervention; and c) the average PMP-SE score at 6 months was statistically significantly higher than at post-intervention. Similarly, for the control group, a) the average PMP-SE score at post-intervention was statistically significantly higher than at pre-intervention; b) the average PMP-SE score at 6 months was statistically significantly higher than at pre-intervention; and c) the average PMP-SE score at 6 months was statistically significantly higher than at post-intervention.

No statistically significant time effects were found for the **Parenting Stress Index (PSI) total scores**, the **PSI Parental Distress** subscale, the **PSI Parent-Child Dysfunctional Interaction** subscale, or for the **PSI Difficult Child** subscale. Most of the effect sizes estimated were considered small, leading to low statistical power and consequently, increasing the likelihood of a type 2 error. Most of the effect sizes were considered small, leading to low statistical power and consequently, increasing the likelihood of a type 2 error.

However, statistically significant time effects for the **Hospital Anxiety and Depression Scale (HADS) total score**, and also for the **HADS Anxiety** and the **HADS Depression** subscales, for both the intervention group and for the control group were found. For all HADS scale and subscales, the scores at 6 months were statistically significantly lower than at pre-intervention, both for the intervention group and the control group. The other comparisons were not statistically significant. Some of the effect sizes were considered small, leading to low statistical power and consequently, increasing the likelihood of a type 2 error.

For the **Infant Behavior Questionnaire Revised (IBQ-R) total score**, a statistically significant time effect was found for the intervention group and also for the control group. For both the intervention and the control group, the average IBQ-R Total score was higher at post-

**Table 3. Within-group differences at pre-intervention, post-intervention, and 6 months.**

| Outcomes Group | Time effect (F, p) | Pre-Post (N, Md), η2, 95%CI, magnitude, power | Pre-6mths (N, Md), η2, 95%CI, magnitude, power | Post-6mths (N, Md), η2, 95%CI, magnitude, power |
|---|---|---|---|---|
| **PMP-SE** | | | | |
| Intervention | F(2,60) = 15.15, p < .001 | Pre:(N = 35, Md = 63) Post:(N = 30, Md = 66), eta2 = 0.195 [0.031, 1] (large), power = 0.793 | Pre:(N = 35, Md = 63) 6mths:(N = 29, Md = 72), eta2 = 0.571 [0.367, 1] (large), power = 1 | Post:(N = 30, Md = 66) 6mths:(N = 29, Md = 72), eta2 = 0.211 [0.036, 1] (large), power = 0.805 |
| Control | F(2,64) = 43.46, p < .001 | Pre:(N = 36, Md = 63.5) Post:(N = 33, Md = 70), eta2 = 0.416 [0.203, 1] (large), power = 0.998 | Pre:(N = 36, Md = 63.5) 6mths:(N = 32, Md = 75.5), eta2 = 0.67 [0.504, 1] (large), power = 1 | Post:(N = 33, Md = 70) 6mths:(N = 32, Md = 75.5), eta2 = 0.502 [0.285, 1] (large), power = 1 |
| **PSI Total** | | | | |
| Intervention | F(2,55) = .12, p = .890 | Pre:(N = 34, Md = 148.5) Post:(N = 29, Md = 148), eta2 = 0.007 [0, 1] (small), power = 0.072 | Pre:(N = 34, Md = 148.5) 6mths:(N = 28, Md = 145), eta2 = 0.004 [0, 1] (small), power = 0.062 | Post:(N = 29, Md = 148) 6mths:(N = 28, Md = 145), eta2 = 0 [0, 1] (small), power = 0.05 |
| Control | F(2,62) = 2, p = .140 | Pre:(N = 34, Md = 155.5) Post:(N = 33, Md = 153), eta2 = 0.103 [0, 1] (medium), power = 0.486 | Pre:(N = 34, Md = 155.5) 6mths:(N = 31, Md = 157), eta2 = 0.001 [0, 1] (small), power = 0.053 | Post:(N = 33, Md = 153) 6mths:(N = 31, Md = 157), eta2 = 0.096 [0, 1] (medium), power = 0.444 |
| **PSI-Parental Distress** | | | | |
| Intervention | F(2,58) = .3, p = .740 | Pre:(N = 36, Md = 48) Post:(N = 30, Md = 46), eta2 = 0.032 [0, 1] (small), power = 0.169 | Pre:(N = 36, Md = 48) 6mths:(N = 28, Md = 48), eta2 = 0.006 [0, 1] (small), power = 0.07 | Post:(N = 30, Md = 46) 6mths:(N = 28, Md = 48), eta2 = 0.001 [0, 1] (small), power = 0.054 |
| Control | F(2,63) = .59, p = .560 | Pre:(N = 35, Md = 47) Post:(N = 33, Md = 49), eta2 = 0.007 [0, 1] (small), power = 0.079 | Pre:(N = 35, Md = 47) 6mths:(N = 31, Md = 51), eta2 = 0.001 [0, 1] (small), power = 0.055 | Post:(N = 33, Md = 49) 6mths:(N = 31, Md = 51), eta2 = 0.053 [0, 1] (small), power = 0.261 |
| **PSI-Parent-Child Dysfunctional Interaction** | | | | |
| Intervention | F(2,54) = .67, p = .520 | Pre:(N = 34, Md = 52.5) Post:(N = 28, Md = 54.5), eta2 = 0.005 [0, 1] (small), power = 0.065 | Pre:(N = 34, Md = 52.5) 6mths:(N = 27, Md = 53), eta2 = 0.045 [0, 1] (small), power = 0.202 | Post:(N = 28, Md = 54.5) 6mths:(N = 27, Md = 53), eta2 = 0.053 [0, 1] (small), power = 0.189 |
| Control | F(2,60) = .65, p = .520 | Pre:(N = 33, Md = 54) Post:(N = 33, Md = 54), eta2 = 0.007 [0, 1] (small), power = 0.074 | Pre:(N = 33, Md = 54) 6mths:(N = 31, Md = 55), eta2 = 0.053 [0, 1] (small), power = 0.239 | Post:(N = 33, Md = 54) 6mths:(N = 31, Md = 55), eta2 = 0.024 [0, 1] (small), power = 0.137 |
| **PSI-Difficult Child** | | | | |
| Intervention | F(2,56) = .2, p = .820 | Pre:(N = 33, Md = 50) Post:(N = 28, Md = 50), eta2 = 0.006 [0, 1] (small), power = 0.068 | Pre:(N = 33, Md = 50) 6mths:(N = 28, Md = 48.5), eta2 = 0.013 [0, 1] (small), power = 0.095 | Post:(N = 28, Md = 50) 6mths:(N = 28, Md = 48.5), eta2 = 0.001 [0, 1] (small), power = 0.052 |
| Control | F(2,54) = 2.17, p = .120 | Pre:(N = 25, Md = 52) Post:(N = 32, Md = 50), eta2 = 0.166 [0.01, 1] (large), power = 0.633 | Pre:(N = 25, Md = 52) 6mths:(N = 29, Md = 52), eta2 = 0.021 [0, 1] (small), power = 0.116 | Post:(N = 32, Md = 50) 6mths:(N = 29, Md = 52), eta2 = 0.053 [0, 1] (small), power = 0.247 |
| **HADS-Total** | | | | |
| Intervention | F(2,59) = 5.84, p < .001 | Pre:(N = 36, Md = 14) Post:(N = 30, Md = 17.5), eta2 = 0.01 [0, 1] (small), power = 0.087 | Pre:(N = 36, Md = 14) 6mths:(N = 29, Md = 12), eta2 = 0.244 [0.057, 1] (large), power = 0.883 | Post:(N = 30, Md = 17.5) 6mths:(N = 29, Md = 12), eta2 = 0.281 [0.067, 1] (large), power = 0.89 |
| Control | F(2,65) = 1.9, p < .001 | Pre:(N = 36, Md = 16.5) Post:(N = 33, Md = 14), eta2 = 0.13 [0.006, 1] (medium), power = 0.606 | Pre:(N = 36, Md = 16.5) 6mths:(N = 32, Md = 9.5), eta2 = 0.361 [0.151, 1] (large), power = 0.991 | Post:(N = 33, Md = 14) 6mths:(N = 32, Md = 9.5), eta2 = 0.212 [0.039, 1] (large), power = 0.826 |
| **HADS-Anxiety** | | | | |
| Intervention | F(2,59) = 4.1, p = .02 | Pre:(N = 36, Md = 9) Post:(N = 30, Md = 9), eta2 = 0.036 [0, 1] (small), power = 0.192 | Pre:(N = 36, Md = 9) 6mths:(N = 29, Md = 6), eta2 = 0.214 [0.039, 1] (large), power = 0.824 | Post:(N = 30, Md = 9) 6mths:(N = 29, Md = 6), eta2 = 0.037 [0, 1] (small), power = 0.171 |

(*Continued*)

**Table 3.** (Continued)

| Outcomes Group | Time effect (F, p) | Pre-Post (N, Md), η2, 95%CI, magnitude, power | Pre-6mths (N, Md), η2, 95%CI, magnitude, power | Post-6mths (N, Md), η2, 95%CI, magnitude, power |
|---|---|---|---|---|
| Control | F(2,64) = 9.43, p < .001 | Pre:(N = 36, Md = 8) Post:(N = 33, Md = 7), eta2 = 0.151 [0.014, 1] (large), power = 0.679 | Pre:(N = 36, Md = 8) 6mths:(N = 32, Md = 6), eta2 = 0.337 [0.13, 1] (large), power = 0.983 | Post:(N = 33, Md = 7) 6mths:(N = 32, Md = 6), eta2 = 0.129 [0.003, 1] (medium), power = 0.568 |
| **HADS-Depression** | | | | |
| Intervention | F(2,58) = 7.65, p < .001 | Pre:(N = 35, Md = 7) Post:(N = 30, Md = 8), eta2 = 0.001 [0, 1] (small), power = 0.053 | Pre:(N = 35, Md = 7) 6mths:(N = 29, Md = 4), eta2 = 0.257 [0.063, 1] (large), power = 0.897 | Post:(N = 30, Md = 8) 6mths:(N = 29, Md = 4), eta2 = 0.339 [0.114, 1] (large), power = 0.962 |
| Control | F(2,65) = 7.29, p < .001 | Pre:(N = 36, Md = 7) Post:(N = 33, Md = 6), eta2 = 0.033 [0, 1] (small), power = 0.188 | Pre:(N = 36, Md = 7) 6mths:(N = 32, Md = 3.5), eta2 = 0.259 [0.072, 1] (large), power = 0.928 | Post:(N = 33, Md = 6) 6mths:(N = 32, Md = 3.5), eta2 = 0.238 [0.054, 1] (large), power = 0.878 |
| **F-PSS-NICU Total** | | | | |
| Intervention | F(2,55) = .12, p = .890 | Pre:(N = 34, Md = 148.5) Post:(N = 29, Md = 148), eta2 = 0.007 [0, 1] (small), power = 0.072 | Pre:(N = 34, Md = 148.5) 6mths:(N = 28, Md = 145), eta2 = 0.004 [0, 1] (small), power = 0.062 | Post:(N = 29, Md = 148) 6mths:(N = 28, Md = 145), eta2 = 0 [0, 1] (small), power = 0.05 |
| Control | F(2,62) = 2, p = .140 | Pre:(N = 34, Md = 155.5) Post:(N = 33, Md = 153), eta2 = 0.103 [0, 1] (medium), power = 0.486 | Pre:(N = 34, Md = 155.5) 6mths:(N = 31, Md = 157), eta2 = 0.001 [0, 1] (small), power = 0.053 | Post:(N = 33, Md = 153) 6mths:(N = 31, Md = 157), eta2 = 0.096 [0, 1] (medium), power = 0.444 |
| **PSS-Visual and Auditive** | | | | |
| Intervention | F(2,58) = 1.77, p = .180 | Pre:(N = 36, Md = 2.67) Post:(N = 29, Md = 3.11), eta2 = 0.079 [0, 1] (medium), power = 0.373 | Pre:(N = 36, Md = 2.67) 6mths:(N = 29, Md = 3), eta2 = 0.088 [0, 1] (medium), power = 0.395 | Post:(N = 29, Md = 3.11) 6mths:(N = 29, Md = 3), eta2 = 0.002 [0, 1] (small), power = 0.055 |
| Control | F(2,63) = 2.2, p = .120 | Pre:(N = 35, Md = 2.78) Post:(N = 33, Md = 2.78), eta2 = 0.067 [0, 1] (medium), power = 0.334 | Pre:(N = 35, Md = 2.78) 6mths:(N = 32, Md = 2.72), eta2 = 0.099 [0, 1] (medium), power = 0.458 | Post:(N = 33, Md = 2.78) 6mths:(N = 32, Md = 2.72), eta2 = 0.01 [0, 1] (small), power = 0.085 |
| **PSS-Baby Behavior** | | | | |
| Intervention | F(2,54) = 1.61, p = .210 | Pre:(N = 36, Md = 2.42) Post:(N = 30, Md = 2.77), eta2 = 0.065 [0, 1] (medium), power = 0.3 | Pre:(N = 36, Md = 2.42) 6mths:(N = 25, Md = 2.62), eta2 = 0.078 [0, 1] (medium), power = 0.309 | Post:(N = 30, Md = 2.77) 6mths:(N = 25, Md = 2.62), eta2 = 0 [0, 1] (small), power = 0.051 |
| Control | F(2,59) = 1.26, p = .290 | Pre:(N = 36, Md = 2.77) Post:(N = 32, Md = 3.08), eta2 = 0.095 [0, 1] (medium), power = 0.451 | Pre:(N = 36, Md = 2.77) 6mths:(N = 27, Md = 2.92), eta2 = 0.016 [0, 1] (small), power = 0.107 | Post:(N = 32, Md = 3.08) 6mths:(N = 27, Md = 2.92), eta2 = 0.001 [0, 1] (small), power = 0.054 |
| **PSS-Parent role** | | | | |
| Intervention | F(2,52) = 2.76, p = .070 | Pre:(N = 35, Md = 2.67) Post:(N = 29, Md = 3.22), eta2 = 0.099 [0, 1] (medium), power = 0.42 | Pre:(N = 35, Md = 2.67) 6mths:(N = 25, Md = 3.11), eta2 = 0.15 [0.002, 1] (large), power = 0.557 | Post:(N = 29, Md = 3.22) 6mths:(N = 25, Md = 3.11), eta2 = 0.003 [0, 1] (small), power = 0.058 |
| Control | F(2,61) = 1.74, p = .180 | Pre:(N = 36, Md = 2.89) Post:(N = 33, Md = 3.33), eta2 = 0.136 [0.007, 1] (medium), power = 0.619 | Pre:(N = 36, Md = 2.89) 6mths:(N = 29, Md = 3.33), eta2 = 0 [0, 1] (small), power = 0.051 | Post:(N = 33, Md = 3.33) 6mths:(N = 29, Md = 3.33), eta2 = 0.057 [0, 1] (small), power = 0.254 |
| **IBQ-R Total** | | | | |
| Intervention | F(2,58) = 15.16, p < .001 | Pre:(N = 36, Md = 1.01) Post:(N = 30, Md = 1.73), eta2 = 0.653 [0.476, 1] (large), power = 1 | Pre:(N = 36, Md = 1.01) 6mths:(N = 27, Md = 4), eta2 = 0.899 [0.837, 1] (large), power = 1 | Post:(N = 30, Md = 1.73) 6mths:(N = 27, Md = 4), eta2 = 0.781 [0.649, 1] (large), power = 1 |
| Control | F(2,63) = 188.02, p < .001 | Pre:(N = 35, Md = 1.08) Post:(N = 32, Md = 2.18), eta2 = 0.615 [0.432, 1] (large), power = 1 | Pre:(N = 35, Md = 1.08) 6mths:(N = 31, Md = 4.35), eta2 = 0.912 [0.861, 1] (large), power = 1 | Post:(N = 32, Md = 2.18) 6mths:(N = 31, Md = 4.35), eta2 = 0.861 [0.778, 1] (large), power = 1 |
| **IBQ-R Surgency** | | | | |
| Intervention | F(2,53) = 207.21, p < .001 | Pre:(N = 34, Md = 0.5) Post:(N = 29, Md = 1.15), eta2 = 0.49 [0.256, 1] (large), power = 0.999 | Pre:(N = 34, Md = 0.5) 6mths:(N = 26, Md = 4.5), eta2 = 0.94 [0.905, 1] (large), power = 1 | Post:(N = 29, Md = 1.15) 6mths:(N = 26, Md = 4.5), eta2 = 0.88 [0.793, 1] (large), power = 1 |

*(Continued)*

**Table 3.** (Continued)

| Outcomes Group | Time effect (F, p) | Pre-Post (N, Md), η2, 95%CI, magnitude, power | Pre-6mths (N, Md), η2, 95%CI, magnitude, power | Post-6mths (N, Md), η2, 95%CI, magnitude, power |
|---|---|---|---|---|
| Control | F(2,60) = 361.46, p < .001 | Pre:(N = 35, Md = 0.46) Post:(N = 32, Md = 1.19), eta2 = 0.53 [0.328, 1] (large), power = 1 | Pre:(N = 35, Md = 0.46) 6mths:(N = 29, Md = 4.62), eta2 = 0.959 [0.935, 1] (large), power = 1 | Post:(N = 32, Md = 1.19) 6mths:(N = 29, Md = 4.62), eta2 = 0.911 [0.855, 1] (large), power = 1 |
| **IBQ-R Negative affect** | | | | |
| Intervention | F(2,57) = 8.7, p < .001 | Pre:(N = 35, Md = 0.5) Post:(N = 29, Md = 1.42), eta2 = 0.574 [0.37, 1] (large), power = 1 | Pre:(N = 35, Md = 0.5) 6mths:(N = 27, Md = 2.58), eta2 = 0.892 [0.826, 1] (large), power = 1 | Post:(N = 29, Md = 1.42) 6mths:(N = 27, Md = 2.58), eta2 = 0.432 [0.196, 1] (large), power = 0.995 |
| Control | F(2,63) = 59.75, p < .001 | Pre:(N = 35, Md = 0.58) Post:(N = 32, Md = 2.17), eta2 = 0.61 [0.425, 1] (large), power = 1 | Pre:(N = 35, Md = 0.58) 6mths:(N = 31, Md = 2.92), eta2 = 0.768 [0.643, 1] (large), power = 1 | Post:(N = 32, Md = 2.17) 6mths:(N = 31, Md = 2.92), eta2 = 0.3 [0.095, 1] (large), power = 0.952 |
| **IBQ-R Effortful Control** | | | | |
| Intervention | F(2,60) = 94.67, p < .001 | Pre:(N = 36, Md = 2) Post:(N = 30, Md = 2.71), eta2 = 0.407 [0.191, 1] (large), power = 0.997 | Pre:(N = 36, Md = 2) 6mths:(N = 27, Md = 5), eta2 = 0.848 [0.763, 1] (large), power = 1 | Post:(N = 30, Md = 2.71) 6mths:(N = 27, Md = 5), eta2 = 0.707 [0.544, 1] (large), power = 1 |
| Control | F(2,63) = 126.78, p < .001 | Pre:(N = 35, Md = 2.25) Post:(N = 32, Md = 3.25), eta2 = 0.443 [0.229, 1] (large), power = 0.999 | Pre:(N = 35, Md = 2.25) 6mths:(N = 31, Md = 5.42), eta2 = 0.866 [0.79, 1] (large), power = 1 | Post:(N = 32, Md = 3.25) 6mths:(N = 31, Md = 5.42), eta2 = 0.826 [0.725, 1] (large), power = 1 |
| **MIBS** | | | | |
| Intervention | F(2,50) = .4, p = .670 | Pre:(N = 32, Md = 2) Post:(N = 29, Md = 2), eta2 = 0.016 [0, 1] (small), power = 0.099 | Pre:(N = 32, Md = 2) 6mths:(N = 26, Md = 1), eta2 = 0.025 [0, 1] (small), power = 0.131 | Post:(N = 29, Md = 2) 6mths:(N = 26, Md = 1), eta2 = 0.021 [0, 1] (small), power = 0.107 |
| Control | F(2,56) = .09, p = .910 | Pre:(N = 33, Md = 2) Post:(N = 31, Md = 2), eta2 = 0 [0, 1] (small), power = 0.05 | Pre:(N = 33, Md = 2) 6mths:(N = 29, Md = 3), eta2 = 0.003 [0, 1] (small), power = 0.06 | Post:(N = 31, Md = 2) 6mths:(N = 29, Md = 3), eta2 = 0.008 [0, 1] (small), power = 0.074 |
| **EPDS** | | | | |
| Intervention | F(2,58) = 5.01, p = .010 | Pre:(N = 36, Md = 11) Post:(N = 30, Md = 9), eta2 = 0.156 [0.013, 1] (large), power = 0.664 | Pre:(N = 36, Md = 11) 6mths:(N = 28, Md = 7.5), eta2 = 0.223 [0.04, 1] (large), power = 0.817 | Post:(N = 30, Md = 9) 6mths:(N = 28, Md = 7.5), eta2 = 0.023 [0, 1] (small), power = 0.123 |
| Control | F(2,64) = 7.18, p < .001 | Pre:(N = 36, Md = 9) Post:(N = 33, Md = 7), eta2 = 0.108 [0, 1] (medium), power = 0.525 | Pre:(N = 36, Md = 9) 6mths:(N = 31, Md = 5), eta2 = 0.332 [0.125, 1] (large), power = 0.98 | Post:(N = 33, Md = 7) 6mths:(N = 31, Md = 5), eta2 = 0.061 [0, 1] (medium), power = 0.291 |

Abbreviations: EPDS: Edinburgh Postnatal Depression Scale; F-PSS-NICU: Parental Stressor Scale: neonatal intensive care unit; HADS: Hospital Anxiety and Depression Scale; IBQ-R: Infant Behavior Questionnaire-Revised Very Short Form; MIBS: Mother-to-Infant Bonding Scale; m-MOS-SS: Modified Medical Outcomes Study Social Support Survey; PMP-SE: Perceived Maternal Self-efficacy; PSI: Parenting Stress Index.

Notes: Time effect was measured using a non-parametric (i.e., Aligned Ranks Transformation) or a parametric (i.e., Linear Mixed Effect) test depending on whether statistically significant covariates were included in the model or not. Effect size interpretation, as suggested by Cohen (1988): small (η2 ≥ 0.01), medium (η2 ≥ 0.06), and large (η2 ≥ 0.14)

intervention than at pre-intervention; higher at 6 months than at pre-intervention; and higher at 6 months than at post-intervention.

The analysis of the **IBQ-R Surgency** showed a statistically significant time effect for the intervention group, and also for the control group. For both the intervention and the control group, the average IBQ-R Surgency score was higher at post-intervention than at pre-intervention; higher at 6 months than at pre-intervention; and higher at 6 months than at post-intervention.

The **IBQ-R Negative Affect** analysis showed a statistically significant time effect for the intervention group, as well as for the control group. For both the intervention and the control group, the average IBQ-R Negative Affect score was: higher at post-intervention than at pre-intervention; higher at 6 months than at pre-intervention; and higher at 6 months than at post-intervention.

The analysis of the **IBQ-R Effortful Control** showed a statistically significant time effect for the intervention group as well as for the control group. For both the intervention and the control group, the average IBQ-R Effortful Control score was higher at post-intervention than at pre-intervention; higher at 6 months than at pre-intervention; and higher at 6 months than at post-intervention.

The results of the **Parental Stressor Scale (PSS) total score** analysis showed a statistically significant time effect for the intervention group but not for the control group. For the intervention group, the average PSS Total score was statistically significantly higher at 6 months than at pre-intervention.

No statistically significant time effects for either the intervention or control groups were found for **PSS Visual and Auditive** and for the **PSS Baby Behavior** subscales. A statistically significant time effect for the **PSS Parent Role** for the intervention group, but not for the control group was found. For the intervention group, the average PSS Parent Role score was statistically significantly higher at post-intervention than at pre-intervention. Most of the effect sizes of these analyses were considered small to medium, leading to low statistical power and consequently, increasing the likelihood of a type 2 error.

The analysis for the **Mother-Infant Bonding Scale (MIBS)** showed no statistically significant time effect for either the intervention or the control groups. All effect sizes of these analyses were considered small, leading to low statistical power and consequently, increasing the likelihood of a type 2 error.

Results for the **Edinburgh Postnatal Depression Scale (EPDS)** showed a statistically significant time effect for the intervention and for the control groups. For both groups, the average EPDS total score was statistically significantly lower at 6 months than at pre-intervention. Some of the effect sizes of these analyses were considered small to medium, leading to low statistical power and consequently, increasing the likelihood of a type 2 error.

The **Questionnaire of satisfaction and acceptability of the intervention** showed high global satisfaction, with 93.8% of the participants being very of extremely satisfied and 81.3% rating the intervention as very or extremely useful (Fig 2). The vast majority (93.8%) would recommend the intervention and found the timing appropriate (97%).

No specific harm or unintended effect was observed in each group.

## Discussion

This randomized controlled trial examined the effects of an early, one-session intervention, the joint observation, provided by an interdisciplinary team in the NICU with a mother and her preterm infant. The primary outcome was the assessment of maternal perceived self-efficacy at one month after study enrollment, assessed with a validated self-report questionnaire. No statistically significant group difference of perceived maternal self-efficacy was observed between the intervention and the control groups at one or six months post-intervention. Similarly, no statistically significant group effects were measured for the secondary outcomes exploring maternal mental health, perception of the parent-infant relationship, and maternal responsiveness. Nonetheless, statistically significant time effects were seen in specific outcomes, including an increase of perceived maternal self-efficacy over time (in both groups), a decrease of psychological distress (HADS total score) (in both groups), an increase of the

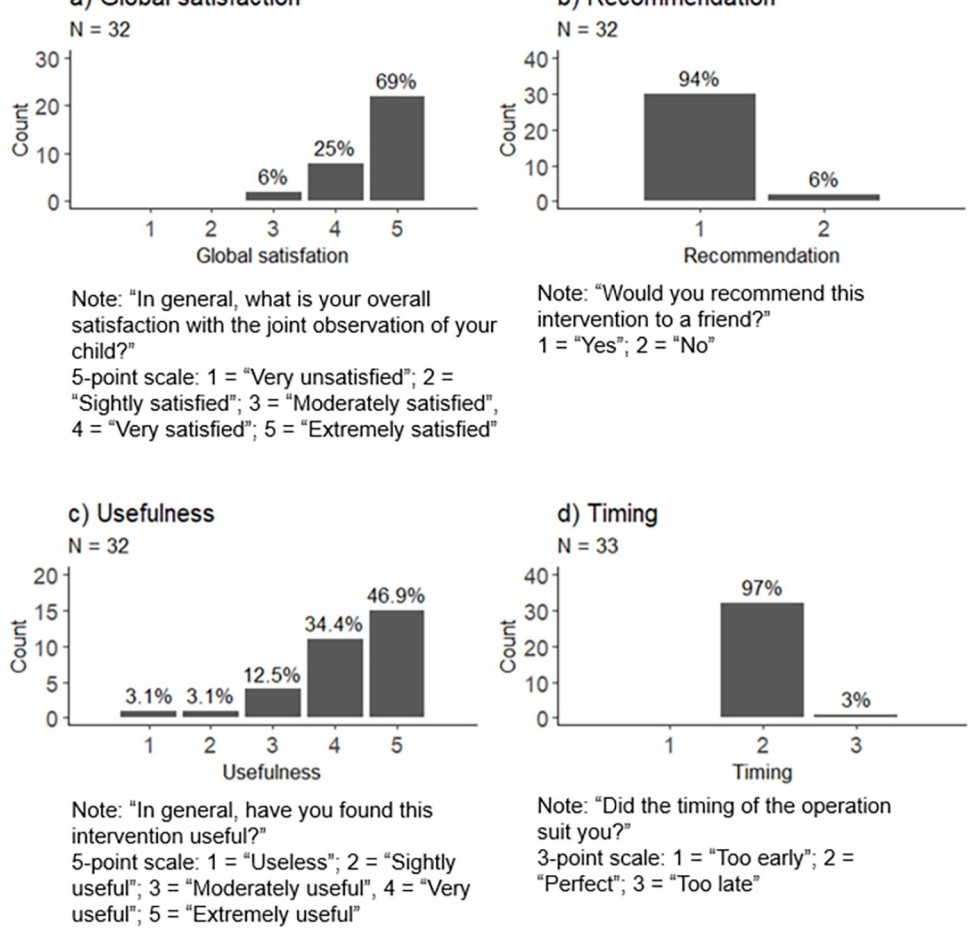

**Fig 2.** Results of the Questionnaire of satisfaction and acceptability of the intervention: a) Global satisfaction, N = 32; b) Recommendation, N = 32; c) Usefulness, N = 32; d) Timing, N = 33.

infant temperament affirmation (Infant Behavior Questionnaire Revised scores) (in both groups), an increase of parental stress related to the parent role in the NICU in the intervention group (Parental Stressor Scale), and a decrease in maternal depression symptoms (EPDS total score) (in both groups). Additionally, mothers declared being very satisfied with the intervention and that they would recommend it to a peer.

In general, demonstrating the effectiveness of an intervention based on the developmental care principles is challenging. It highly relies on the intervention design, the dose-effect, multiple confounding factors, and reliable outcome measures. Nevertheless, over the last two decades, several interventions have proved having an impact on specific outcomes [8]. Among others, the *Newborn Individualized Developmental Care and Assessment Program* (NIDCAP) [48], the *Assessment of Preterm Infant Behavior* (APIB) [49], the *Creating Opportunities for Parent Empowerment* (COPE) [50], and *the Family Nurture Intervention* (FNI) [51] addressed similar aims as our intervention, including promoting parent-infant relationship, empowerment of parents, and reduction of parental psychological distress. Positive effects were reported on specific short-and long term outcomes toward the infant, such as shorter length of stay [50, 52], decreased problem behavior in childhood [48], and improved neurodevelopmental scores [53], although sustained effect could not be necessarily demonstrated [52]. Improved

parental outcomes were related to more sensitive interactions and communication with the infant [48, 50, 51], better knowledge of the preterm behavior [49], lower stress, anxiety and depression symptoms [54], and increased maternal sensitivity. These interventions were generally part of a comprehensive developmental program, were administered regularly during the NICU stay and often continued after home discharge; their design thus differed from ours. Nevertheless, the JOIN observation, as part of an intervention program, had in a previous study demonstrated to have an effect on maternal and child outcomes in our centre, including lower maternal posttraumatic stress symptoms and increased sensitivity, as well as improved infant cooperation during interactions at 4 months of age [25]. In addition, the JOIN observation combines elements originating from four distinct theories of infant development, namely the method described by Brazelton et Nugent [26], the synactive model of Als [27], the sensori-motor approach of Bullinger [28] and the interactive guidance [29, 30]. Given that there was evidence for each element to promote either neonatal support or parental sensitivity, we were reasonably confident that incorporating them to build this early, one-session intervention in the NICU would be effective.

## Limitations

Despite an appropriate randomized controlled study design and a suitable sample size, our study did not show group differences in regards to maternal perceived self-efficacy and other outcomes examining mental health and parent-infant relationship. Possible reasons, which might be regarded as limitations of the study, are related to 1) the infants' and mothers' socio-demographic characteristics, 2) the high standard of routine care, 3) the study design, 4) the intervention providers, 5) the choice of the primary outcome, and 6) the sample size. First, the targeted preterm population was rather homogeneous in terms of gestational age at birth and considered as low-risk regarding severity of neonatal complications. In most cases, the parents reported a high socio-economic status, with a third having a university degree, no migrant status, and speaking French, which may constitute a selection bias. Included mothers also reported good social support, as indicated by the high m-MOS-SS scores in both groups, which might have subsequently decreased the risk of psychological distress. Eligibility to participate in the study was not based on perceived mother-infant vulnerability, but on clinical criteria (gestational age, cardio-pulmonary stability of the preterm neonate,. . .). Low variability in maternal demographics and characteristics may have influenced the results toward a statistically non-significant group difference. Second, the standard care in our unit (provided to the control group) includes a comprehensive developmental care program, as well as an effective psychological support, provided to approximately 2/3 of the families of preterm born infants below 32 weeks gestation in a sustained or infrequent manner, according to the parents' needs. This high level of supportive measures in standard care may have reduced the potential additional benefits of the JOIN intervention. Thirdly, although mothers were satisfied with the intervention, it may not be well adapted to their needs. In particular, providing the intervention just once may be insufficient and it is possible that there would be a measurable dose-response effect with repeated interventions in the NICU or even provided in the home environment once the baby is discharged. Furthermore, the mothers who benefited from the intervention in the NICU might be sensitized and more alert toward their baby's behaviour and they might not have the psychological follow-up needed to reinforce their competences. Forth, the multidisciplinary team providing the intervention was comprised of nurses in the NICU, psychologists, and child psychiatrists and was thus heterogeneous in terms of background and qualification. Yet, a structured training and regular supervision provided by experienced psychologists/child psychiatrists ensured the fidelity of the intervention.

Nevertheless, this may have influenced the quality of the intervention with the mothers. Fifth, the joint intervention previously proved to have an impact on maternal sensitivity [25], and the present study was designed according to this finding assuming that perceived maternal self-efficacy would encompass this competence and others, including receptive parental behaviour. Thus, the primary outcome could perhaps have directly measured mothers' interactive behaviour instead. Sixth, despite adequate sample size of the two groups based on the initial power calculation, we might also hypothesize that the absence of an effect of the intervention was related to the low number of participants, especially given that smaller effect sizes than expected were observed. The sample size calculation was performed on previous published means related to perceived parental self-efficacy in a population of mothers of term [18] and preterm [31] neonates, which might proved not to have been entirely suitable. We therefore recommend that future research is carried out on a larger sample size assuming smaller effect sizes. Finally, a bias may have been introduced due to the unblinding of the participants and clinicians, as well as a possible contamination due to improvement of usual care by healthcare providers.

The degree of maternal satisfaction with the intervention is encouraging, as are the statistically significant time-effects we observed in several outcome measures, including maternal perceived self-efficacy, as well as anxiety and depression symptoms. The high-quality routine care implemented in the NICU to support families as described above may have effectively supported mothers during the hospitalisation of their preterm neonate and may have helped to overcome difficulties through self-empowerment, regardless of the intervention. Responses on the IBQ-R questionnaire revealed increasing total and subscale scores in both groups over time, reflecting a natural evolution of the child's temperament over the first months of life. Interestingly, the temperament factors explored by this questionnaire have been associated with the perception of parental closeness, satisfaction, competence and child's language development [55]. In addition, a statistically significantly higher score was observed on the Parental Stressor Scale in the intervention group at 6 months post-intervention compared to pre-intervention, which was mainly driven by the subscale score exploring the *Parent Role*. This finding was not expected but we may hypothesize that the mothers who benefited from the intervention were sensitized to enhance their parental role, which may have caused a form of stress in parenting in the NICU or once the infant is discharged home [34].

Although this study did not find an effect of the intervention on perceived maternal self-efficacy, it does not mean that no association would have been found in a different context given the limitations that have been raised above, especially the sample size, the choice of a mildly vulnerable population or the single intervention design. To support this assumption, we observed that the experience of providing or receiving the intervention was felt to be positive from the maternal side, as well as the multidisciplinary team. Families appeared to be grateful for the time invested by the professionals to provide this highly specialized moment of care. It may ultimately promote the therapeutic alliance, the reliance, and the adherence to the global care project, while building the partnership between the parents and the medical team according to the family-centered care principles. Within the multidisciplinary team, the JOIN intervention allowed to create a positive dynamic, where each member contributed their specific expertise and thus helped to promote the establishment of the developmental care in the NICU.

## Perspectives

Future research may helpfully investigate the effect of this intervention when offered to parents of preterm infants experiencing particularly high levels of psychological distress and thus at

risk of developing mental health problems. Identifying these parents could be based on screening parents regarding their psychological vulnerability or distress whilst their infant is still hospitalised in the NICU, for instance by using specific questionnaires, such as the m-MOS-SS, EPDS or HADS. Similarly, the sickest infants with neonatal complications bringing vital risks should be identified, as their parents may experience more mental health difficulties related to this critical situation. Exploring the effect of the intervention in a higher risk population would be of great interest, hypothesizing that vulnerable families may benefit the most from this type of support. We may need to adapt the setting of the intervention to better adjust to the parents' needs and the baby's maturation. Hence, previous interventions that demonstrated positive effects on mother-infant outcomes provided more sustained support, at least for a 4-week period [50, 51] or at several time-points from term equivalent age to 4 months of corrected age, while the infant has already been discharged [25]. More individualized parenting support adapted to the parental and infant needs in terms of duration, frequency and content might be developed too, focusing on the more vulnerable population. Moreover, as many interventions so far have focused on maternal mental health, it would be essential to include fathers/co-parents to study their own perceptions, as healthcare partners within an inclusive family-centered care unit.

## Conclusions

The joint observation (JOIN), an early one-session, brief intervention provided in the NICU with the mother-infant dyad was not ensued with improved perceived maternal self-efficacy at one and six months post-intervention, nor with effects on maternal mental health and parent-infant relationship perception, compared to controls. However, this study highlighted positive changes over time in several outcomes explored through questionnaires on perceived self-efficacy, stress, depression, anxiety, and child's temperament perception, reflecting the high-level psychological support and an environment fostering neurodevelopment in the NICU. This intervention is part of the supportive measures that can be offered to parents and their preterm infant during a vulnerable period, following the family-centered care principles.

## Supporting information

**S1 Fig. Perceived Maternal Parenting Self-Efficacy–Within-group scores distributions.**
Within-group scores distributions are separately displayed for the intervention group and for the control group, with statistically significant time effects in both groups. PMP-SE: Perceived Maternal Parenting Self-Efficacy; 1-Pre: Pre-intervention's time point; 2-Post: Post-intervention's time point; 6-Mths: 6 months' time points.
(DOCX)

**S1 Table. Missing values in questionnaires.**
(DOCX)

**S2 Table. Between-group differences at baseline–Outcomes.** The results of the maternal questionnaires at baseline (pre-intervention) for both the intervention and the control groups are presented in the table showing no statistically significant between-group differences.
(DOCX)

**S3 Table. Cronbach alphas for study questionnaires.**
(DOCX)

## Acknowledgments

### The JOIN Research Consortium

Lead author: Antje Horsch (Department of Woman-Mother-Child, Clinic of Neonatology, Lausanne University Hospital Center and University of Lausanne, Lausanne, Switzerland), antje.horsch@chuv.ch

Laureline Besuchet (Department of Woman-Mother-Child, Clinic of Neonatology, Lausanne University Hospital Center and University of Lausanne, Lausanne, Switzerland), Cindy Boche (Department of Woman-Mother-Child, Clinic of Neonatology, Lausanne University Hospital Center and University of Lausanne, Lausanne, Switzerland), Ayala Borghini (Department of Child and Adolescent Psychiatry, Lausanne University Hospital Center and University of Lausanne, Lausanne, Switzerland), Josée Despars (Department of Child and Adolescent Psychiatry, Lausanne University Hospital Center and University of Lausanne, Lausanne, Switzerland), Alice Manser Chenaux (Department of Woman-Mother-Child, Clinic of Neonatology, Lausanne University Hospital Center and University of Lausanne, Lausanne, Switzerland), Noémie Faure (Centre Sages-Femmes, Vevey, Switzerland and UniVers Famille, Châtel-St-Denis, Switzerland), Valérie Goyer (Department of Woman-Mother-Child, Clinic of Neonatology, Lausanne University Hospital Center and University of Lausanne, Lausanne, Switzerland), Aurélie Le Berre (Department of Woman-Mother-Child, Clinic of Neonatology, Lausanne University Hospital Center and University of Lausanne, Lausanne, Switzerland), Maryline Monnier (Department of Woman-Mother-Child, Clinic of Neonatology, Lausanne University Hospital Center and University of Lausanne, Lausanne, Switzerland), Mathilde Morisod Harari (Department of Child and Adolescent Psychiatry, Lausanne University Hospital Center and University of Lausanne, Laupower sanne, Switzerland), Roxane Romon (Department of Woman-Mother-Child, Clinic of Neonatology, Lausanne University Hospital Center and University of Lausanne, Lausanne, Switzerland), Juliane Schneider (Department of Woman-Mother-Child, Clinic of Neonatology, Lausanne University Hospital Center and University of Lausanne, Lausanne, Switzerland), Catherine Sperandio (Department of Woman-Mother-Child, Clinic of Neonatology, Lausanne University Hospital Center and University of Lausanne, Lausanne, Switzerland), Chloé Tenthorey (Department of Woman-Mother-Child, Clinic of Neonatology, Lausanne University Hospital Center and University of Lausanne, Lausanne, Switzerland), Jean-François Tolsa (Department of Woman-Mother-Child, Clinic of Neonatology, Lausanne University Hospital Center and University of Lausanne, Lausanne, Switzerland), and Aline Yersin (Department of Child and Adolescent Psychiatry, Lausanne University Hospital Center and University of Lausanne, Lausanne, Switzerland).

We would like to thank Lyne Jaunin, Geneviève Métrailler Dizi, and Manon Macherel for contributing to the writing of the ethics proposal. We acknowledge the contribution of Carole Muller-Nix and Margot Forcada-Guex to the development of the intervention. We are also grateful to Priska Udriot, Joanne Horisberger, Cassie Pernet, Stéphanie Huguelet, and Vania Sandoz for the help with data collection. We would like to acknowledge the Clinic of Neonatology, and Carole Richard, in particular for her support. Finally, we would like to thank the parents for their participation.

## Author Contributions

**Conceptualization:** Mathilde Morisod Harari, Noémie Faure, Ayala Borghini, Jean-François Tolsa, Antje Horsch.

**Data curation:** Juliane Schneider, Mathilde Morisod Harari, Antje Horsch.

**Formal analysis:** Juliane Schneider, Alain Lacroix, Antje Horsch.

**Methodology:** Mathilde Morisod Harari, Noémie Faure, Ayala Borghini, Antje Horsch.

**Project administration:** Juliane Schneider, Jean-François Tolsa, Antje Horsch.

**Supervision:** Juliane Schneider, Mathilde Morisod Harari, Antje Horsch.

**Writing – original draft:** Juliane Schneider, Noémie Faure, Alain Lacroix, Antje Horsch.

**Writing – review & editing:** Mathilde Morisod Harari, Ayala Borghini, Jean-François Tolsa, Antje Horsch.

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
