## [Decision Letter · Decision Letter 0]

6 Sep 2023

PONE-D-23-00084Joint observation in NICU (JOIN): a randomized controlled trial testing an early, one-session intervention during preterm care to improve perceived maternal self-efficacy and other mental health outcomesPLOS ONE

Dear Dr. Schneider,

Thank you for submitting your manuscript to PLOS ONE. After careful consideration, we feel that it has merit but does not fully meet PLOS ONE’s publication criteria as it currently stands. Therefore, we invite you to submit a revised version of the manuscript that addresses the points raised during the review process.

The manuscript has been evaluated by two reviewers, and their comments are available below.

The reviewers have raised a number of concerns, many of which relate to the methods and analysis.

Could you please carefully revise the manuscript to address all comments raised?

We look forward to receiving your revised manuscript.

Kind regards,

Steve Zimmerman, PhD

Associate Editor, PLOS ONE

3. One of the noted authors is a group or consortium [JOIN Research Consortium]. In addition to naming the author group, please list the individual authors and affiliations within this group in the acknowledgments section of your manuscript. Please also indicate clearly a lead author for this group along with a contact email address.

4. Please upload a new copy of Figure 2 as the detail is not clear. Please follow the link for more information: https://blogs.plos.org/plos/2019/06/looking-good-tips-for-creating-your-plos-figures-graphics/" https://blogs.plos.org/plos/2019/06/looking-good-tips-for-creating-your-plos-figures-graphics/

Reviewers' comments:

Reviewer's Responses to Questions

**Comments to the Author**

1. Is the manuscript technically sound, and do the data support the conclusions?

Reviewer #1: Yes

Reviewer #2: Yes

2. Has the statistical analysis been performed appropriately and rigorously? 

Reviewer #1: I Don't Know

Reviewer #2: I Don't Know

3. Have the authors made all data underlying the findings in their manuscript fully available?

Reviewer #1: Yes

Reviewer #2: Yes

4. Is the manuscript presented in an intelligible fashion and written in standard English?

Reviewer #1: Yes

Reviewer #2: Yes

5. Review Comments to the Author

Reviewer #1: Important note: This review pertains only to ‘statistical aspects’ of the study and so ‘clinical aspects’ [like medical importance, relevance of the study, ‘clinical significance and implication(s)’ of the whole study, etc.] are to be evaluated [should be assessed] separately/independently. Further please note that any ‘statistical review’ is generally done under the assumption that (such) study specific methodological [as well as execution] issues are perfectly taken care of by the investigator(s). This review is not an exception to that and so does not cover clinical aspects {however, seldom comments are made only if those issues are intimately / scientifically related & intermingle with ‘statistical aspects’ of the study}. Agreed that ‘statistical methods’ are used as just tools here, however, they are vital part of methodology [and so should be given due importance]. I look at the manuscript in/with statistical view point, other reviewer(s) look(s) at it with different angle so that in totality the review is very comprehensive. However, there should be efforts from authors side to improve (may be by taking clues from reviewer’s comments). Therefore, please do not limit the revision only (with respect) to comments made here.

COMMENTS: Although there are few appreciable points of this manuscript [example: application of Aligned Ranks Transform (ART) tests], I have few serious concerns {mainly regarding sample size and ‘Statistical analysis’ section}. First, refer to lines 128-129 where it is stated that “Based on an initial power calculation [24], a sample size of 68 participants was planned”, however, remember that since this is an independent publication [and not one in series], you are supposed to give some (more) details. Estimated number may be correct (but generally number per arm/group is reported/yielded). However, note that even quoted reference (24) says “it is planned that 80 mothers will be enrolled to anticipate possible participant withdrawal” but not mentioned “what is ‘anticipate possible participant withdrawal’ {i.e., drop-out} rate?” According to reference quoted, estimation of required sample size is based on sample of preterm (M=58.51, SD=12.57) and of term-born (M=65.9, SD=8.2) which implies that effect size assumed is much more than actually observed [refer to table-2]. Are the controls full term-born infants in the Neonatal Intensive Care Unit (NICU) environment? According to reference 24 (‘Control group’ section) ‘Participants in the control group will receive treatment as-usual’. That means they are also pre-term born infants in the Neonatal Intensive Care Unit (NICU) environment {which is very correct indeed}. What is true? The question (then) is, ‘are the assumptions made for estimation process, correct?’ Please give details of ‘control group’ selection & constitution. Can ‘control’ group definition be different for sample size estimation and for trial execution? Please mind you that this is a scientific/academic document and so all details should be clearly/correctly communicated (do not take readers’ for granted).

How the required minimum sample size for this study was determined is nevertheless a very-very important question [one of the important items in CONSORT checklist, item 7a] for any type of study (clinical trial or else) which needs to be discussed in adequate details {including assumptions made at the time of estimation, (expected) power of the study, software used, etc.}. In line 120 it is stated that “This manuscript follows the CONSORT guidelines.”. This implies that you know well that while reporting findings from [and even planning] ‘Clinical Trial’ one should follow CONSORT guidelines. Unfortunately, though it is said, even important items {like How sample size was determined (Item 7a), Random Sequence generation (Item 8a), Allocation concealment (Item 9), Blinding (Item 11a)} of/in CONSORT checklist are not found [since your article type is ‘Clinical Trial’, you are supposed to cover (with adequate detailing) these items in the report or even in ‘Protocol’]. Only stating (lines 376-8) that “a bias may have been introduced due to the unblinding of the participants and clinicians, as well as a possible contamination due to improvement of usual care by healthcare providers” is not sufficient. You have to give explanation as to why any sort of blinding was not possible / not done. For every required point, you are not supposed to indicate (just refer to) the protocol (though published). Mind you that this is an independent publication [and not one in series] as said earlier.

Note that according to table-2 on page 158 of Cohen’s paper “A power primer” in Psychological Bulletin, 1992, vol.:112, pp 155-159 [which is a sort of summary of the excellent book by Jacob Cohen entitled ‘Statistical power analysis for the behavioral sciences’, Academic Press, 1977, New York] even for medium effect size you need n=64 per group and for small effect size it is as large as 393 per group (type-I error=0.05, power=80%). Kindly note that ‘n’ is per group/arm & not for entire study/trial.

I request authors to read the following text [particularly in context of column 4 (P-values column) of Table 1 {‘Between-group differences at baseline - Maternal and infant sociodemographic variables’}] pasted from one famous standard textbook on ‘Medical Research Methodology’:

To provide a description of baseline characteristics is entirely reasonable (since it is clearly important in assessing to whom the results of the trial can be applied), however, statistical comparison of baseline characteristics when random allocation/assignment is used/done [often for good/standard/leading journals these days] is not required, because even if P-value(s) turn(s) out to be significant (while comparing baseline characteristics despite random allocation), it is, by definition, a false positive as you then are supposed to be testing ‘randomization’ then, which in any single trial may not balance all baseline characteristics (particularly when sample sizes are small). Remember that ‘randomization’ is a sort of ‘insurance’ and not a guarantee scheme. Authors may please refer to following articles:

References:

1. Stuart J. Pocock, et al., ‘Subgroup analysis, covariate adjustment and baseline comparisons in clinical trial reporting: current practice and problems’, Statistics in medicine, 2002; 21:2917–2930 [Particularly page 2927]

2. Harrington D, et al., ‘New guidelines for statistical reporting in the journal’, N Engl J Med 2019;381:285-6

[Important message (indirectly/ultimately indicated) from these articles: Never do any comparison with respect to ‘baseline’ characteristics {by applying statistical significance test(s)}, when allocation is done randomly].

However, Statistical comparison [only with respect to important/indicated variables] of baseline characteristics may be performed, to find out if analysis adjustment (say stratified analyses or else) is required with respect to these variables.

Note {referring to Table-1 with respect to Education; Migrant; Marital status; Number of children and Largo score categories}: There are few limitations/conditions [like more than 80% cells should have expected cell frequency more or equal to 5 as well as no cell frequency should be ‘zero’] of Chi-square test to keep in mind.

Though the measures/tools used are appropriate [examples: Primary outcome - Perceived Maternal Parenting Self-efficacy (PMP-SE); Secondary outcomes - Parental Stressor Scale : neonatal intensive care unit (F-PSS-NICU); Parenting Stress Index – Short Form (PSI - SF); Hospital Anxiety and Depression Scale (HADS); Edinburgh Postnatal Depression Scale (EPDS); etc.], most of them yield data that are in [at the most] ‘ordinal’ level of measurement [and not in ratio level of measurement for sure {as the score two times higher does not indicate presence of that parameter/phenomenon as double (for example, a Visual Analogue Scales VAS score or say ‘depression’ score)}]. This is given here because as in line 206-7 “We checked the equivalence between groups on social demographics and outcome measures at baseline using t-tests (and Wilcoxon rank sum tests)”, then the application of suitable non-parametric test(s) is/are indicated/advisable [even if distribution may be ‘Gaussian’ (also called ‘normal’)]. Agreed that there is/are no non-parametric test(s)/technique(s) available to be used as alternative in all situation(s) [suitable / most desired/applicable], but should be used whenever/wherever they are available. Good that at few places Wilcoxon rank sum tests are/were used (where?). However, in short use suitable non-parametric test(s)/technique(s) while dealing with data that are in ‘ordinal’ level of measurement even if distribution may be ‘Gaussian’.

By referring to lines 394-6 [Although this study did not find an effect of the intervention on perceived maternal self-efficacy, the experience of providing or receiving it was felt to be positive from the maternal side, as well as the multidisciplinary team.], I would like to point-out one fact. Remember that “Absence of evidence is not evidence of absence” [Altman DG, Bland JM. BMJ volume 311, 1995, p 485 (Reprinted: Australian Veterinary Journal 1996;74, 311)]. {Even when P-value is not significantly lower that is null hypothesis of no difference / no association is not rejected, (in short, result is not significant), that does not amount to evidence of absence i.e., it does not imply that there no difference / no association. It only implies that there is no (i.e., these samples do not provide) [say enough] evidence to prove (rather indicate with certain specified confidence level) the difference / association}. Therefore, conclusion(s) from any study [in which result(s) is/are not significant], should be drawn in the light of this fact. But would like to also note the second/later part that ‘it was felt to be positive from the maternal side, as well as the multidisciplinary team’. It may lead to doubt that “whether this ‘non-significance’ is resulted because of less than required sample size of the study [wrong control group used]?”

Moreover, limitations (if any) of the study are not mentioned/listed anywhere. Does that mean {according to authors} there are none? As pointed out in ‘important note’ above “This review pertains only to ‘statistical aspects’ of the study and so ‘clinical aspects’ should be assessed separately/independently [one should carefully consider/look at the clinical implications of the study]. In my opinion, to rescue this article (which is not quite impossible), some amount of re-vision (re-drafting) may be needed. However, please do not limit the revision only (with respect) to comments made here.

Reviewer #2: This is a very well written paper on an important subject. NICU parents and infants are a vulnerable population and it is critical to develop interventions that assist them in establishing more optimal relationships to support infant development and parental mental health. Ideal interventions are feasible to disseminate in NICUs with varying degrees of psychosocial support and can provide support for parents and infants from varying backgrounds. JOIN is possibly one such intervention, and therefore, research into its effectiveness in improving parental self-efficacy and parental and infant mental health and developmental outcomes is very important. The authors designed this RCT based on previous studies on JOIN and used self-report questionnaires to assess multiple important outcomes. Ultimately, they do not find a meaningful difference between the two arms of the study, but they go on to describe potential reasons for this lack of a difference in a very thoughtful way. Overall, I believe this is an important research in the field and lays the foundation for future inquiries into this intervention.

Here are a few minor questions/suggested revisions:

Abstract:

- line 25: what do they mean by "adjusted parent-infant relationship", consider a different adjective.

Introduction:

- lines 82-86: Please further elaborate on how parental self-efficacy is distinguished from confidence or competence.

- line 106: instead of "by the intervention", consider "In this intervention"

Methods"

-line 118: instead of the "protocol of the", consider "the protocol for the"

- it would be helpful to more clearly divide the exclusion criteria to maternal factors and infant factors. Consider putting ":" after each rather than ";".

-line 125: instead of "aged", consider "age"

-line 141: instead of "this moment", consider "this period"

-line 163: what aspect of the infant?

- in the abstract it is mentioned that authors measure PTSD outcomes as well, but the included measurements only look at stress, anxiety and depression, please clarify this.

- for all the outcome measures used, please consider commenting on their psychosocial properties for the NICU population

Discussion:

The discussion is particularly strong and well written. It would be helpful if authors explain why they chose a one session intervention (question of the dosing of the intervention) and briefly tell us how they envision adjusting the intervention to each unique NICU dyad's needs.

Their discussion about the role of sociodemographic factors in their population (high education level, etc), less medical acuity, possibility of directly measuring mother's interactive behaviors rather than their self report of their self-efficacy, and the strength of their treatment as usual (good developmental care program and strong resources for psychological support on the unit) are all very well thought out.

I very much enjoyed this paper and look forward to seeing further research on JOIN!

6. PLOS authors have the option to publish the peer review history of their article (what does this mean?). If published, this will include your full peer review and any attached files.

Reviewer #1: No

Reviewer #2: No

---

## [Author Response · Author response to Decision Letter 0]

3 Nov 2023

Response: The manuscript has been checked and changes have been made according to the journal’s style requirements. 

Response: The initial project was submitted to the ethics committee of the “Canton de Vaud” in December 2015. At that time, there was no requirement from the ethical board to mention the possibility of publicly sharing data in the information document to the participants. Thus, participants did not consent formally to share their data, even if de-identified. In that context, we are not allowed to publish or upload an anonymized data set in a repository. However, we could share the data upon reasonable request to the corresponding author. We mentioned this point in the cover letter. 

3. One of the noted authors is a group or consortium [JOIN Research Consortium]. In addition to naming the author group, please list the individual authors and affiliations within this group in the acknowledgments section of your manuscript. Please also indicate clearly a lead author for this group along with a contact email address.

Response: The individual authors from the JOIN Research Consortium have been listed ant their affiliations checked. The lead author (Pr. Antje Horsch) has been identified in the acknowledgements section (Page 35, Line 515).

4. Please upload a new copy of Figure 2 as the detail is not clear. Please follow the link for more information: https://blogs.plos.org/plos/2019/06/looking-good-tips-for-creating-your-plos-figures-graphics/" https://blogs.plos.org/plos/2019/06/looking-good-tips-for-creating-your-plos-figures-graphics/

Response: A new version of the Figure 2 has been uploaded, with better resolution and higher font size of the legends. We used the PACE software to ensure our figure meets the requirements. 

Response: All the references have been checked.

Reviewers' comments:

Reviewer's Responses to Questions

Comments to the Author

1. Is the manuscript technically sound, and do the data support the conclusions?

Reviewer #1: Yes

Reviewer #2: Yes

2. Has the statistical analysis been performed appropriately and rigorously? 

Reviewer #1: I Don't Know

Reviewer #2: I Don't Know

3. Have the authors made all data underlying the findings in their manuscript fully available?

Reviewer #1: Yes

Reviewer #2: Yes

4. Is the manuscript presented in an intelligible fashion and written in standard English?

Reviewer #1: Yes

Reviewer #2: Yes

5. Review Comments to the Author

 

Reviewer #1: Important note: This review pertains only to ‘statistical aspects’ of the study and so ‘clinical aspects’ [like medical importance, relevance of the study, ‘clinical significance and implication(s)’ of the whole study, etc.] are to be evaluated [should be assessed] separately/independently. Further please note that any ‘statistical review’ is generally done under the assumption that (such) study specific methodological [as well as execution] issues are perfectly taken care of by the investigator(s). This review is not an exception to that and so does not cover clinical aspects {however, seldom comments are made only if those issues are intimately / scientifically related & intermingle with ‘statistical aspects’ of the study}. Agreed that ‘statistical methods’ are used as just tools here, however, they are vital part of methodology [and so should be given due importance]. I look at the manuscript in/with statistical view point, other reviewer(s) look(s) at it with different angle so that in totality the review is very comprehensive. However, there should be efforts from authors side to improve (may be by taking clues from reviewer’s comments). Therefore, please do not limit the revision only (with respect) to comments made here.

We thank the reviewer for his/her comments on the statistical part, which constitutes a very important component of the study. Issues regarding the clinical aspects have been addressed according to the comments of the second reviewer. 

You will find below our point-to-point response to the comments. Please note that the page and line numbers to which the changes refer correspond to the clean version of the revised manuscript. We also added a separate bibliography with the references that were added or mentioned at the end of the present document. 

COMMENTS: Although there are few appreciable points of this manuscript [example: application of Aligned Ranks Transform (ART) tests], I have few serious concerns {mainly regarding sample size and ‘Statistical analysis’ section}. First, refer to lines 128-129 where it is stated that “Based on an initial power calculation [24], a sample size of 68 participants was planned”, however, remember that since this is an independent publication [and not one in series], you are supposed to give some (more) details. Estimated number may be correct (but generally number per arm/group is reported/yielded). However, note that even quoted reference (24) says “it is planned that 80 mothers will be enrolled to anticipate possible participant withdrawal” but not mentioned “what is ‘anticipate possible participant withdrawal’ {i.e., drop-out} rate?” According to reference quoted, estimation of required sample size is based on sample of preterm (M=58.51, SD=12.57) and of term-born (M=65.9, SD=8.2) which implies that effect size assumed is much more than actually observed [refer to table-2]. 

Response: We agree with the reviewer that we should have better explained the power calculation in the present manuscript, and we must not take for granted the fact that the study protocol has been previously published. In that sense, we added a paragraph in the “Method” section detailing the sample size calculation, as follows (Pages 7-8, Lines 140-148) : “The power calculation was based on previous publications measuring perceived parental self-efficacy in parents of term (Leahy-Warren, McCarthy, & Corcoran, 2012) and preterm (Barnes & Adamson-Macedo, 2007) neonates. As no previous study compared this specific outcome after a similar intervention in two groups of mothers of preterm neonates in the NICU, we made the assumption that the mothers of preterm neonates benefitting from the JOIN intervention will report comparable perceived self-efficacy as mothers of term neonates. Using the G*Power software (Faul, Erdfelder, Lang, & Buchner, 2007) allowing sample size determination and according to the published means and SD in these two samples (term: M=65.9, SD=8.2; preterm: M=58.1, SD 12.57), we needed to recruit 68 participants (α=0.05, 1-β=0.80, unilateral hypothesis). To anticipate possible withdrawal, we planned to enroll 80 mother-infant dyads in the present study”. 

Secondarily, we acknowledge that the expected effect sizes were smaller than expected in the present study and we clarified this in the “Limitations” paragraph with this sentence (Page 32, Line 447-450) : “Sixth, despite adequate sample size of the two groups based on the initial power calculation, we might also hypothesize that the absence of an effect of the intervention was related to the low number of participants, especially given that smaller effect sizes than expected were observed”.

Are the controls full term-born infants in the Neonatal Intensive Care Unit (NICU) environment? According to reference 24 (‘Control group’ section) ‘Participants in the control group will receive treatment as-usual’. That means they are also pre-term born infants in the Neonatal Intensive Care Unit (NICU) environment {which is very correct indeed}. What is true? The question (then) is, ‘are the assumptions made for estimation process, correct?’ Please give details of ‘control group’ selection & constitution. Can ‘control’ group definition be different for sample size estimation and for trial execution? Please mind you that this is a scientific/academic document and so all details should be clearly/correctly communicated (do not take readers’ for granted).

Response: We thank the reviewer for pointing this out and agree that there might a bit of confusion in the “Method” section regarding the control group. The control group comprises the same inclusion criteria as the intervention group, i.e. mothers of preterm neonates born between 28 and 32 6/7 weeks of gestational age. We acknowledge that the power calculation was based on previous publications on term neonates and that we made the assumption that mothers of preterm neonates in our control group might be comparable to mothers of term neonates in that study (Leahy-Warren et al., 2012) when reporting perceived self-efficacy. We did so because no previous study comparing perceived self-efficacy after a similar intervention as ours was available. Thus, we specified again in the “Trial design, procedure, data collection and timing” paragraph that both groups concerned mothers of preterm neonates, as follows (Page 8, Lines 150-153) : “The recruitment of mothers of preterm neonates was performed by research nurses who approached the eligible participants once their infants were stable enough to avoid disturbance during a critical period and ensure their emotional availability”.

We also specified in the “Sample size” calculation paragraph the following sentence explaining the rationale of the power calculation (Page 7, Lines 141-144) : “As no previous study compared this specific outcome after a similar intervention in two groups of mothers of preterm neonates in the NICU, we assumed that the mothers of preterm neonates benefitting from the JOIN intervention will report comparable perceived self-efficacy as mothers of term neonates”.

How the required minimum sample size for this study was determined is nevertheless a very-very important question [one of the important items in CONSORT checklist, item 7a] for any type of study (clinical trial or else) which needs to be discussed in adequate details {including assumptions made at the time of estimation, (expected) power of the study, software used, etc.}. In line 120 it is stated that “This manuscript follows the CONSORT guidelines.”. This implies that you know well that while reporting findings from [and even planning] ‘Clinical Trial’ one should follow CONSORT guidelines. Unfortunately, though it is said, even important items {like How sample size was determined (Item 7a), Random Sequence generation (Item 8a), Allocation concealment (Item 9), Blinding (Item 11a)} of/in CONSORT checklist are not found [since your article type is ‘Clinical Trial’, you are supposed to cover (with adequate detailing) these items in the report or even in ‘Protocol’]. Only stating (lines 376-8) that “a bias may have been introduced due to the unblinding of the participants and clinicians, as well as a possible contamination due to improvement of usual care by healthcare providers” is not sufficient. You have to give explanation as to why any sort of blinding was not possible / not done. For every required point, you are not supposed to indicate (just refer to) the protocol (though published). Mind you that this is an independent publication [and not one in series] as said earlier.

Response: We agree with the reviewer that we referred a lot to our previous publication of the study protocol and that the present manuscript lacks details, especially regarding the CONSORT guidelines requirements. We completed the “Sample size calculation” paragraph by specifying the type of software we used for the sample size determination, as follows (Pages 7-8, Lines 144-147) : “Using the G*Power software(Faul et al., 2007) allowing sample size determination and according to the published means and SD in these two samples (term: M=65.9, SD=8.2; preterm: M=58.1, SD 12.57), we needed to recruit 68 participants (α=0.05, 1-β=0.80, unilateral hypothesis)”. 

We also made changes accordingly in the “Trial design, procedure, data collection and timing” paragraph by adding information on random sequence generation, allocation concealment, blinding and procedures in each group. Modifications were the following (Pages 8-9, Lines 150-172) : “This monocentric RCT aimed to test an intervention compared with treatment-as-usual. The recruitment of mo

---

## [Decision Letter · Decision Letter 1]

10 Jan 2024

PONE-D-23-00084R1Joint observation in NICU (JOIN): a randomized controlled trial testing an early, one-session intervention during preterm care to improve perceived maternal self-efficacy and other mental health outcomesPLOS ONE

Dear Dr. Schneider,

Thank you for submitting your manuscript to PLOS ONE. After careful consideration, we feel that it has merit but does not fully meet PLOS ONE’s publication criteria as it currently stands. Therefore, we invite you to submit a revised version of the manuscript that addresses the points raised during the review process.

We look forward to receiving your revised manuscript.

Kind regards,

Astawus Alemayehu Feleke, Ph.D Can..

Academic Editor

PLOS ONE

Journal Requirements:

Additional Editor Comments:

I would like to suggest the authors to address the concerns raised by reviewer 1.

Reviewers' comments:

Reviewer's Responses to Questions

**Comments to the Author**

1. If the authors have adequately addressed your comments raised in a previous round of review and you feel that this manuscript is now acceptable for publication, you may indicate that here to bypass the “Comments to the Author” section, enter your conflict of interest statement in the “Confidential to Editor” section, and submit your "Accept" recommendation.

Reviewer #1: (No Response)

Reviewer #2: All comments have been addressed

2. Is the manuscript technically sound, and do the data support the conclusions?

Reviewer #1: (No Response)

Reviewer #2: Yes

3. Has the statistical analysis been performed appropriately and rigorously? 

Reviewer #1: (No Response)

Reviewer #2: I Don't Know

4. Have the authors made all data underlying the findings in their manuscript fully available?

Reviewer #1: (No Response)

Reviewer #2: No

5. Is the manuscript presented in an intelligible fashion and written in standard English?

Reviewer #1: (No Response)

Reviewer #2: Yes

6. Review Comments to the Author

Reviewer #1: COMMENTS: Although all the comments are answered and few positively attended, frankly speaking I am not very satisfied or convinced about few of the answers/responses [I said ‘FEW of the answers/responses’ about which I am not very satisfied, however, most other responses are alright]. Though, I do not have any specific recommendation, only as system requirement I am choosing ‘major revision’ [assuming that the respected editor would like to give chance to authors for improvement of the manuscript]. Nevertheless, few changes [example: adding (Page 32, Lines 447-452) “Sixth, despite adequate sample size of the two groups based on the initial power calculation, we might also hypothesize that the absence of an effect of the intervention was related to the low number of participants, especially given that smaller effect sizes than expected were observed. The sample size calculation was performed on previous published means related to perceived parental self-efficacy in a population of mothers of term (Leahy-Warren et al., 2012) and preterm (Barnes & Adamson-Macedo, 2007) neonates, which might prove not to have been entirely suitable] are appreciable but let the respected editor decide the future course.

Reviewer #2: (No Response)

7. PLOS authors have the option to publish the peer review history of their article (what does this mean?). If published, this will include your full peer review and any attached files.

Reviewer #1: No

Reviewer #2: No

---

## [Author Response · Author response to Decision Letter 1]

30 Jan 2024

Journal Requirements:

Response: The reference list has been checked, according to the Journal requirements.

Additional Editor Comments:

I would like to suggest the authors to address the concerns raised by reviewer 1.

Response: Please see below our response to the comments raised by the reviewer 1. 

Reviewers' comments:

Reviewer's Responses to Questions

Comments to the Author

1. If the authors have adequately addressed your comments raised in a previous round of review and you feel that this manuscript is now acceptable for publication, you may indicate that here to bypass the “Comments to the Author” section, enter your conflict of interest statement in the “Confidential to Editor” section, and submit your "Accept" recommendation.

Reviewer #1: (No Response)

Reviewer #2: All comments have been addressed

2. Is the manuscript technically sound, and do the data support the conclusions?

Reviewer #1: (No Response)

Reviewer #2: Yes

3. Has the statistical analysis been performed appropriately and rigorously? 

Reviewer #1: (No Response)

Reviewer #2: I Don't Know

4. Have the authors made all data underlying the findings in their manuscript fully available?

Reviewer #1: (No Response)

Reviewer #2: No

5. Is the manuscript presented in an intelligible fashion and written in standard English?

Reviewer #1: (No Response)

Reviewer #2: Yes

6. Review Comments to the Author

Reviewer #1: COMMENTS: Although all the comments are answered and few positively attended, frankly speaking I am not very satisfied or convinced about few of the answers/responses [I said ‘FEW of the answers/responses’ about which I am not very satisfied, however, most other responses are alright]. Though, I do not have any specific recommendation, only as system requirement I am choosing ‘major revision’ [assuming that the respected editor would like to give chance to authors for improvement of the manuscript]. Nevertheless, few changes [example: adding (Page 32, Lines 447-452) “Sixth, despite adequate sample size of the two groups based on the initial power calculation, we might also hypothesize that the absence of an effect of the intervention was related to the low number of participants, especially given that smaller effect sizes than expected were observed. The sample size calculation was performed on previous published means related to perceived parental self-efficacy in a population of mothers of term (Leahy-Warren et al., 2012) and preterm (Barnes & Adamson-Macedo, 2007) neonates, which might prove not to have been entirely suitable] are appreciable but let the respected editor decide the future course.

Response: We appreciate the time you took to look again at our manuscript, and we thank you for acknowledging the changes we made to improve the manuscript’s content. 

We agree that the limitations need to be carefully acknowledged in this study and they have been listed in the corresponding section pages 31 to 34. Regarding the mention you made about the sample size, we acknowledge that our original power calculation assumed larger effect sizes than we were able to detect in our study. Thus, we added the following statement allowing to better envision future studies (page 32, lines 452-453): “We therefore recommend that future research is carried out on a larger sample size assuming smaller effect sizes.”

Reviewer #2: (No Response)

7. PLOS authors have the option to publish the peer review history of their article (what does this mean?). If published, this will include your full peer review and any attached files.

Do you want your identity to be public for this peer review? For information about this choice, including consent withdrawal, please see our Privacy Policy.

Reviewer #1: No

Reviewer #2: No

---

## [Editor Report · Decision Letter 2]

19 Mar 2024

Joint observation in NICU (JOIN): a randomized controlled trial testing an early, one-session intervention during preterm care to improve perceived maternal self-efficacy and other mental health outcomes

PONE-D-23-00084R2

Dear Dr. Schneider,

We’re pleased to inform you that your manuscript has been judged scientifically suitable for publication and will be formally accepted for publication once it meets all outstanding technical requirements.

Kind regards,

Astawus Alemayehu Feleke, Ph.D Can..

Academic Editor

PLOS ONE

Additional Editor Comments (optional):

I would like to congratulate the authors for this valuable and insightful research paper which can contribute important information to the field. the authors addressed all comments, suggestions, and concerns raised by the reviewer and Now, the paper meets the publication criteria for PLOS ONE journal, so I propose this manuscript to be published in PLOS ONE journal.